# Task-Gated Multi-Expert Collaboration Network for Degraded Multi-Modal Image Fusion

Yiming Sun [1]  Xin Li [1]  Pengfei Zhu [1]  Qinghua Hu [2 3 4]  Dongwei Ren [2 3 4]  Huiying Xu [5]  Xinzhong Zhu [5]

## Abstract

Multi-modal image fusion aims to integrate complementary information from different modalities to enhance perceptual capabilities in applications such as rescue and security. However, real-world imaging often suffers from degradation issues, such as noise, blur, and haze in visible imaging, as well as stripe noise in infrared imaging, which significantly degrades model performance. To address these challenges, we propose a task-gated multi-expert collaboration network (TG-ECNet) for degraded multi-modal image fusion. The core of our model lies in the task-aware gating and multi-expert collaborative framework, where the task-aware gating operates in two stages: degradation-aware gating dynamically allocates expert groups for restoration based on degradation types, and fusion-aware gating guides feature integration across modalities to balance information retention between fusion and restoration tasks. To achieve this, we design a two-stage training strategy that unifies the learning of restoration and fusion tasks. This strategy resolves the inherent conflict in information processing between the two tasks, enabling all-in-one multi-modal image restoration and fusion. Experimental results demonstrate that TG-ECNet significantly enhances fusion performance under diverse complex degradation conditions and improves robustness in downstream applications. The code is available at https://github.com/LeeX54946/TG-ECNet.

[1]School of Automation, Southeast University, Nanjing, China [2]College of Intelligence and Computing, Tianjin University, Tianjin, China [3]Engineering Research Center of City Intelligence and Digital Governance, Ministry of Education of the People's Republic of China, Tianjin, China [4]Haihe Lab of ITAI, Tianjin, China [5]School of Computer Science and Technology, Zhejiang Normal University, Jinhua, China. Correspondence to: Pengfei Zhu <zhupengfei@tju.edu.cn>.

*Proceedings of the $42^{nd}$ International Conference on Machine Learning*, Vancouver, Canada. PMLR 267, 2025. Copyright 2025 by the author(s).

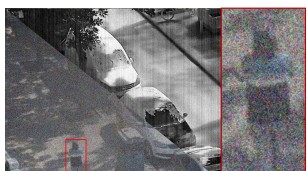

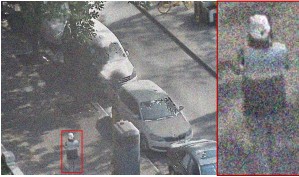
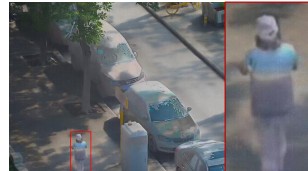

| Degraded multi-modal image | DRMF (ACM MM 2024) |
|---|---|
| Text-IF (CVPR 2024) | Fused image (Ours) |

*Figure 1.* Comparison of degraded multi-modal image fusion. For multi-degraded multi-modal images, the proposed method shows superior restoration and fusion quality compared to the SOTA methods DRMF (Tang et al., 2024) and Text-IF (Yi et al., 2024), especially in the highlighted regions, where our method better preserves details and reduces noise.

## 1. Introduction

Multi-modal image fusion plays a pivotal role in applications (Sun et al., 2022a; Liu et al., 2024) such as rescue, security, and surveillance by integrating complementary information from diverse imaging modalities (Ma et al., 2019a; Zhang et al., 2021; Xu et al., 2022a; Liu et al., 2021; 2022b). The visible modality provides superior textural details, while the infrared modality effectively highlights thermal targets. The combination of visible and infrared imaging advantages enables the synthesis of fused images suitable for all-day operation (Sun et al., 2022b; Liu et al., 2023; Zhao et al., 2023b; Cao et al., 2023; Liu et al., 2020), thereby enhancing application robustness in complex environments.

However, real-world multi-modal imaging systems are susceptible to performance degradation caused by multiple interfering factors (Tang et al., 2022b; 2023; Sun et al., 2024; Tang et al., 2024; Yi et al., 2024). Specifically, the key component for capturing thermal information in infrared sensors is the focal plane array, whose variations in the bias voltage of the readout circuit in different columns often result in

alternating bright and dark stripe noise in the infrared image. Visible imaging is also susceptible to mixed degradation by noise, blur, haze, and other disturbances. These degradations seriously weaken the visual quality of multi-modal images (as shown in Fig. 1), leading to a degradation in the performance of the fused images and affecting their robustness and effectiveness in critical downstream applications.

To address the problem of degraded multi-modal image fusion, a direct approach is to cascade two tasks sequentially: first, restore the degraded images, and then fusion the restored multi-modal images. However, this strategy presents dual limitations. On one hand, the diversity of degradation types necessitates pre-storing a large number of restoration models for multiple degradation conditions, entailing prohibitive costs when addressing complex scenarios (Tang et al., 2022b; 2023; Sun et al., 2024). On the other hand, the inherent disconnection between these two tasks leads to performance degradation: restoration operations aiming to restore degraded information may inadvertently weaken features beneficial for fusion, while fusion processes risk propagating restoration failures into the final results. Current state-of-the-art methods, such as DRMF (Tang et al., 2024) and Text-IF (Yi et al., 2024), attempt to address these limitations by integrating restoration and fusion into a unified framework through diffusion models or text-guided mechanisms. Nevertheless, as illustrated in Fig. 1, these approaches demonstrate insufficient capability in handling complex multi-degradation scenarios, yielding suboptimal fusion outcomes that hinder practical applications in real-world environments with intricate degradation conditions.

In this work, we propose the task-gated multi-expert collaboration network (TG-ECNet) that unifies degraded multi-modal image restoration and fusion into a common framework. Specifically, we propose task-aware gating modules for the degradation and fusion stages, respectively. Among them, degradation-aware gating ensures robust restoration based on input features by adaptively learning the degradation types (*e.g.*, noise, blur, haze, streak noise) and selecting the most appropriate processing paths with a group of experts. Meanwhile, the fusion-aware gating selectively aggregates multi-modal features and selects the most valuable complementary information for high-quality fusion by multi-expert collaboration. TG-ECNet adopts a two-stage training strategy to bridge the image restoration and fusion tasks, which ensures a balanced optimization of the restoration and fusion objectives. By decoupling the learning process, the two-stage strategy minimizes the interference between the tasks and ultimately achieves all-in-one multi-modal image restoration and fusion. This unified approach not only improves the fusion quality but also enhances the adaptability of the model to downstream applications in various complex degradation scenarios. In addition, we also construct a large-scale benchmark, DeMMI-RF, for degraded image restoration and fusion. The main contributions of this paper are summarized as follows:

- We propose a unified framework for degraded multi-modal image restoration and fusion, which bridges different tasks together through a two-stage training strategy to learn inter-task information while avoiding mutual interference, enabling all-in-one processing.

- We propose the task-aware gating and multi-expert collaboration module. The degradation-aware gating adapts to different degradation types and selects the optimal expert group for image restoration, while the fusion-aware gating dynamically balances the information retention between fusion and restoration tasks to achieve better fusion performance.

- We construct a large-scale degraded multi-modal image fusion benchmark, DeMMI-RF, which contains more than $30,000$ multi-modal data of different degradation types, including those from UAVs and driving viewpoints. Results on multiple datasets validate the superior performance of the model in complex degraded scenarios and robustness for downstream applications.

## 2. Related Work

### 2.1. Multi-modal Image Fusion

Multi-modal image fusion integrates complementary information from various modalities (*e.g.*, visible and infrared) to produce enriched representations (Ma et al., 2019a; Liu et al., 2024). Deep learning has driven progress in this field, with CNNs (Zhang et al., 2020; Wang et al., 2022a; Sun et al., 2022b; Xu et al., 2022b) and GANs (Liu et al., 2022a; Ma et al., 2019b) learning fusion rules directly from the data. Recent innovations, such as attention mechanisms and transformer-based architectures (Tang et al., 2022c; Wang et al., 2022b), enhance performance by modeling long-range dependencies and modality-specific features. Frameworks like MGDN (Guan et al., 2023) further unify sub-tasks within fusion. However, most existing deep learning methods focus solely on fusion and neglect the impact of degradations, such as noise, blur, and haze, which are common in real-world scenarios. This limitation significantly reduces their robustness and applicability in practical applications.

### 2.2. Degraded Image Restoration

Degraded image restoration aims to recover high-quality images from degraded inputs, addressing issues like noise, blur, and haze. While traditional approaches (Xia et al., 2023) target individual degradations, real-world scenarios often involve intertwined degradations. All-in-one image restoration aims to address diverse degradation types using a unified model. Recent advances leverage specific task

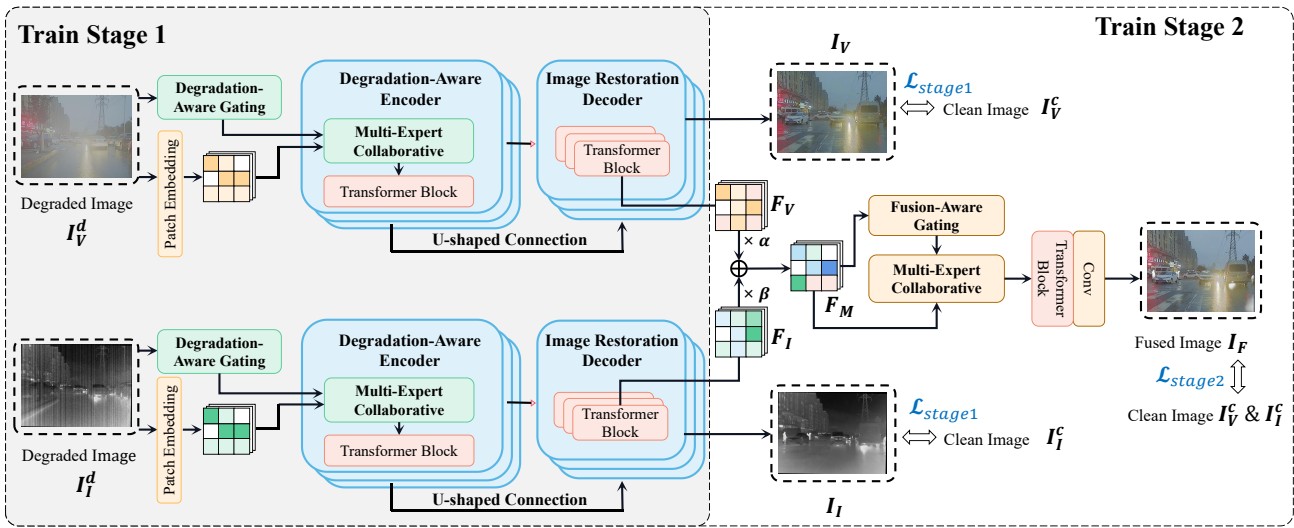

*Figure 2.* The architecture of TG-ECNet. TG-ECNet consists of a U-shape Transformer framework, a degradation-aware gating, a fusion-aware gating, and corresponding multi-expert collaboration framework.

learning (Zamir et al., 2022) or degradation-aware mechanisms (Li et al., 2022; Potlapalli et al., 2023; Cui et al., 2025) to dynamically adapt to varying degradation types without requiring prior knowledge of the degradation. Most existing methods are limited to single-modal images or address restoration and fusion separately, failing to handle multi-modal data effectively. While recent works (Tang et al., 2024; Li et al., 2024a) tackle quality issues (*e.g.*, low light, haze, and noise) in multi-modal fusion, they remain restricted to specific degradations. Some approaches, such as Text-IF (Yi et al., 2024) and Text-DiFuse (Zhang et al., 2024), employ text-guided restoration but require prior degradation knowledge, while others (*e.g.*, AWFusion (Li et al., 2024b)) are confined to weather-related scenarios. Methods like (Tang et al., 2025) suffer from complex architectures and limited datasets, hindering optimal performance. In contrast, our TG-ECNet introduces a unified framework that jointly optimizes restoration and fusion, ensuring robust and high-quality fusion in real-world scenarios.

## 3. Methods

### 3.1. Overall Architecture

In this work, we propose the task-gated multi-expert collaboration network (TG-ECNet) to address the challenges of image quality degradation in multi-modal image fusion through an all-in-one approach. The framework contains: the U-shape Transformer (Zamir et al., 2022) for feature extraction and image decoding, a degradation-aware gated multi-expert collaborative module for all-in-one image restoration, and a fusion-aware gated multi-modal collaboration module for adaptive image fusion. Additionally, a two-stage training strategy is employed to balance the

learning of the restoration and fusion tasks.

The architecture of TG-ECNet is shown in Fig. 2. We send a pair of degraded infrared image $I_I^d \in \mathbb{R}^{H \times W \times 1}$ and degraded visible image $I_V^d \in \mathbb{R}^{H \times W \times 3}$ into the patch embedding and degradation-aware gating to extract the features. Then these features are fed into the degradation-aware encoder and image restoration decoder to get $I_I$ and $I_V$. The structure of the encoders and decoders follows (Zamir et al., 2022). It should be noted that the infrared and visible modal branches share encoder and decoder weights. In the second stage, the features $F_I$ and $F_V$ from the decoder are also fed into the fusion branch to obtain $I_F$. The overall structure of our model can be notated as:

$$I_F = TG\text{-}ECNet(I_V^d, I_I^d). \quad (1)$$

### 3.2. Task-Aware Gating and Multi-Expert Collaborative

As shown in Fig 3, the task-aware gating and multi-expert collaborative is the core component of TG-ECNet, integrating two key mechanisms to facilitate adaptive restoration and fusion. The first mechanism, degradation-aware gating, is implemented in the U-shape Transformer encoder to dynamically identify degradation types present in the input images, such as noise, blur, haze in visible images, and stripe noise in infrared images. Based on the degradation types, task-specific prompts are generated to guide the selection of appropriate processing pathways, allowing the model to adaptively restore images tailored to their degradation types. The second mechanism, fusion-aware gating, is applied in the image fusion stage and focuses on selectively aggregating features from multiple modalities, such as visible and infrared images. By weighting the contributions from each modality based on their relevance and complementarity, fusion-aware gating ensures that the most

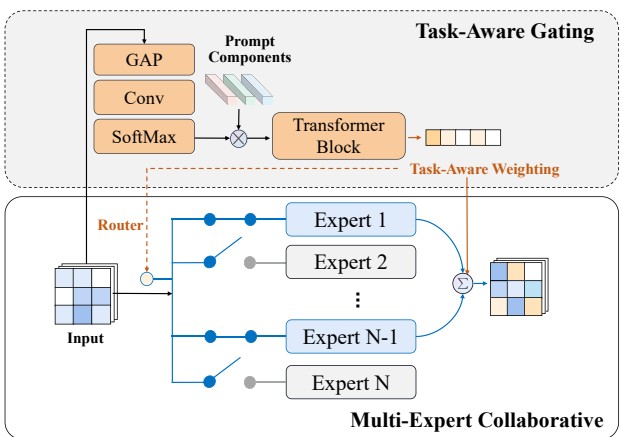

*Figure 3.* Task-aware gating and multi-expert collaborative.

informative features are combined, resulting in high-quality fusion output. This mechanism ensures effective feature integration, enabling robust multi-modal fusion that preserves essential information from both input sources.

**Degradation-Aware Gated Image Restoration.** In the first training stage, TG-ECNet restores degraded images, beginning with a degraded image $I_V^d$ or $I_I^d$. The model applies the degradation-aware gating mechanism $G_{Degrad}(I_V^d)$ to adjust based on the specific degradation present in the input image. This gating enables the model to dynamically adapt to different degradation types, ensuring that the appropriate processing strategy is applied. The input image is then processed by the degradation-aware encoder, which utilizes a multi-expert collaborative network $\{E_1^{Degrad}, ..., E_N^{Degrad}\}$, including a transformer block, to extract and refine features that are specific to the encountered degradation types. The multi-expert mechanism selects the most relevant expert to learn different image features, optimizing the restoration process. When processing the visible image, the formalization is as follows,

$$Output = \sum_{i=1}^{N} G_{Degrad}(I_V^d)_i \cdot E_i^{Degrad}(I_V^d). \quad (2)$$

After feature extraction, the decoder, composed of transformer blocks, generates a restored version of the image $I_V$, which is compared with the clean ground truth image $I_V^c$ to compute the loss, guiding the model to improve restoration accuracy. The same restoration process is applied to the infrared degraded image $I_I^d$, generating a restored infrared image $I_I$.

**Fusion-Aware Gated Image Fusion.** In the second training stage, TG-ECNet focuses on fusing the restored visible and infrared images to create a high-quality fused output. The inputs to this stage are the restored visible feature $F_V$ and infrared feature $F_I$ from the previous stage. We introduce learnable weight parameters $\alpha$ and $\beta$ to guide the fusion

of different modal features: $F_M = \alpha \cdot F_V + \beta \cdot F_I$. The fusion-aware gating mechanism $G_{Fus}(F_M)$ is applied to the decoder to enhance the fusion process by focusing on the most relevant features from both modalities. This gating mechanism enables the model to selectively weight the contributions of each modality, ensuring that the most important information from both images is retained in the fused result. The model then uses the multi-expert collaborative network $\{E_1^{Fus}, ..., E_N^{Fus}\}$ to combine features from both visible and infrared images.

$$Output_{Fus} = \sum_{i=1}^{N} G_{Fus}(F_M)_i \cdot E_i^{Fus}(F_M). \quad (3)$$

This collaborative fusion approach ensures that features from both modalities are effectively integrated, preserving critical details from both sources. The final fused image $I_F$ is obtained by merging these features, and a loss is calculated by comparing the fused image with the clean ground truth $I_V^c$ and $I_I^c$. This loss is used to refine the model's fusion capabilities, ensuring high-quality fusion results that combine the best features of both input modalities.

### 3.3. Training Strategy and Loss Function

To balance restoration and fusion tasks and minimize interference during optimization, we employ a two-stage training strategy. The model uses a combination of loss functions during training. In Stage 1 (Restoration-focused training), the network is first trained to address degradations in individual modalities, using loss functions tailored to specific restoration tasks (*e.g.*, denoising, deblurring). The restoration loss in Stage 1 is calculated by comparing the restored images $I_V$ and $I_I$ with their respective clean ground truth images $I_V^c$ and $I_I^c$. In Stage 2, the fusion loss is computed by comparing the fused image $I_F$ with the clean fused ground truth image. The formulation of the loss function($\mathcal{L}_{grad}^{Degrad}$, and $\mathcal{L}_{load}^{Degrad}$) follows (Cao et al., 2023).

$$\mathcal{L}_{res} = \|I_V - I_V^c\| + \|I_I - I_I^c\|. \quad (4)$$

$$\mathcal{L}_{stage1} = \mathcal{L}_{res} + \mathcal{L}_{grad}^{Degrad} + \mathcal{L}_{load}^{Degrad}. \quad (5)$$

In Stage 2 (Fusion-focused training), after restoration, the network is fine-tuned for fusion tasks. The final loss for the entire network is the sum of the restoration losses from Stage 1 and the fusion loss from Stage 2, allowing the model to simultaneously optimize both tasks. By decoupling the learning processes, the strategy ensures that restoration and fusion tasks do not interfere with each other, ultimately enabling all-in-one image restoration and fusion. The formulation of the fusion loss function($\mathcal{L}_{pixel}$, $\mathcal{L}_{grad}$, and $\mathcal{L}_{load}^{Fus}$) follows (Cao et al., 2023).

$$\mathcal{L}_{stage2} = \mathcal{L}_{pixel} + \mathcal{L}_{grad} + \mathcal{L}_{load}^{Fus}. \quad (6)$$

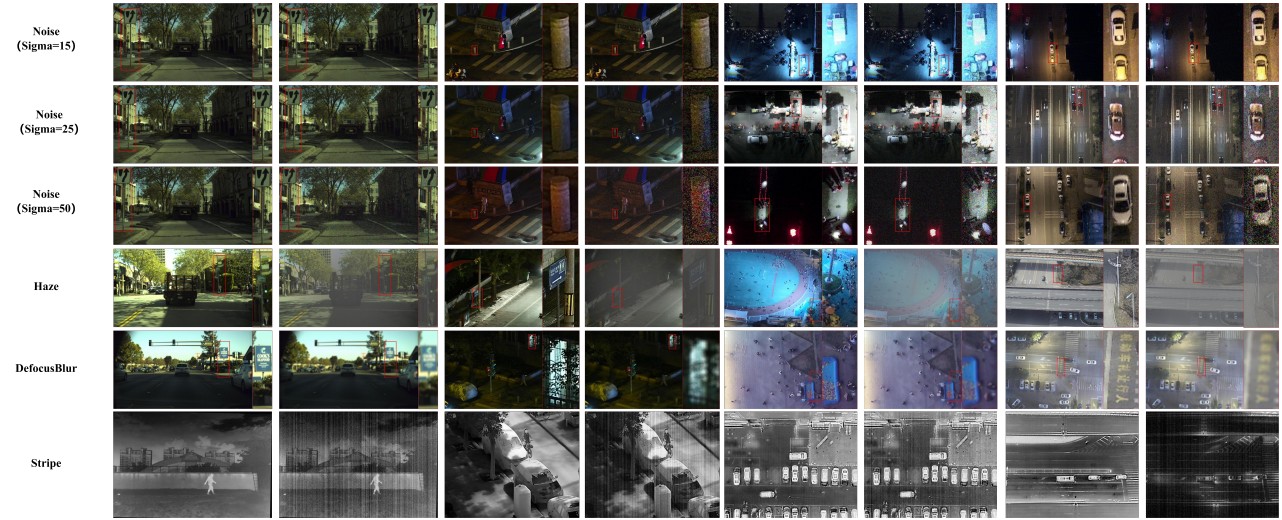

*Figure 4.* The visualization of the proposed DeMMI-RF dataset.

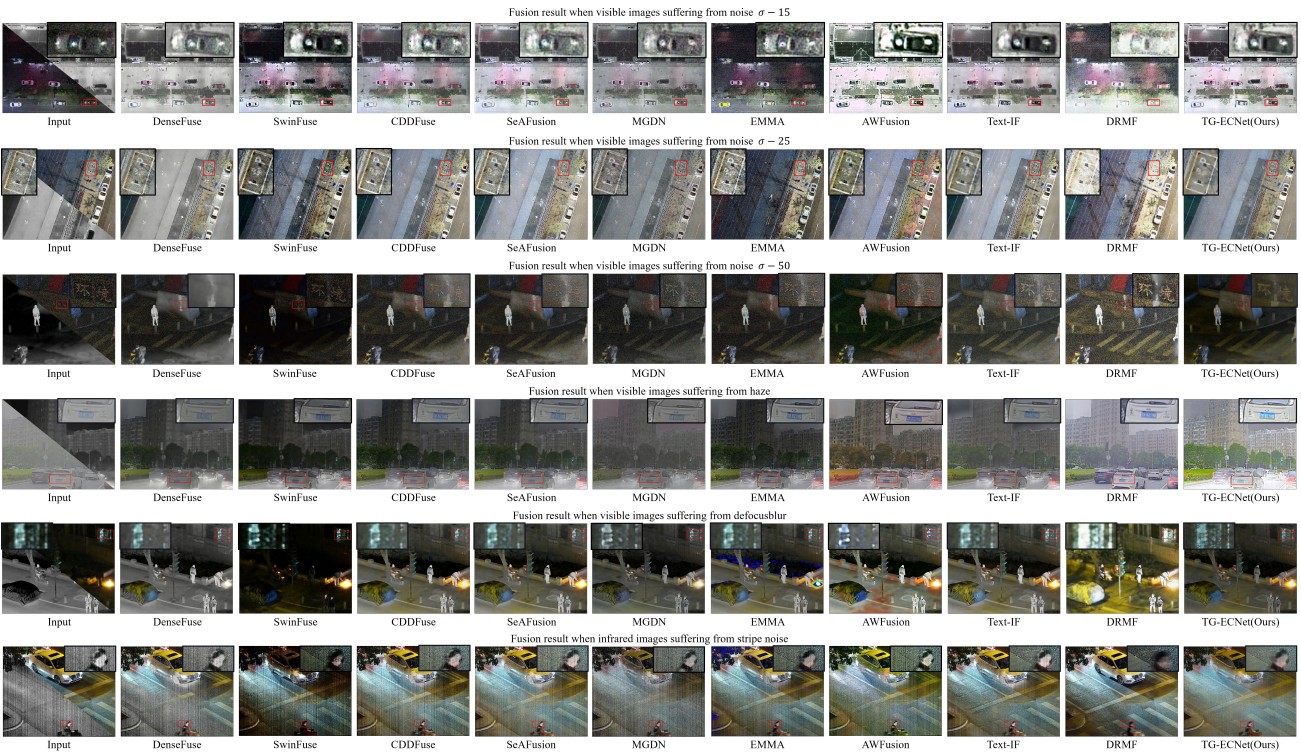

*Figure 5.* Qualitative comparisons of various methods.

## 4. Experiment

### 4.1. Experimental Setting

**Implementation Details.** All experiments in this paper were performed on 6 NVIDIA GeForce RTX 4090 GPUs, and the model was implemented using the PyTorch 1.12.0 framework. During the training phase, we used the Adam optimizer to optimize the network, setting the initial learning rate to $1.0 \times 10^{-4}$ and adjusting it using the cosine annealing strategy. In addition, we randomly cropped the images to a size of $128 \times 128$ pixels for training. In each small batch, data augmentation was performed by flipping the images horizontally or vertically to expand the training sample size. We conduct experiments on both our constructed dataset and the EMS dataset (Yi et al., 2024). We trained a single model under 6 degradation settings. The first stage training process lasted for 30 epochs, and the model was directly tested across multiple restoration tasks. The second stage training process lasted for 30 epochs, and the model was directly tested across multiple restoration and fusion tasks. In experiments, the number of experts $N$ and the number of selected experts $K$ were heuristically set to 11 and 6, respectively.

Relevant analysis can be found in Appendix A.3.

**Details of the Constructed Dataset.** Existing infrared-visible image datasets rarely take complex degradation scenarios into account, and there are no multimodal degradation datasets related to low-altitude drone perception scenarios. Therefore, we constructed a multimodal multi-degradation image dataset, DeMMI-RF, involving urban street view perspectives and low-altitude drone perspectives. Our DeMMI-RF dataset includes 6 types of degradation: high/medium/low levels of Gaussian noise, haze, defocus blur, and striped noise. Typical cases of the dataset are shown in Fig. 4, which includes both ground and drone scenarios. DeMMI-RF has 26631 training datasets and 9895 testing datasets, providing a powerful benchmark for degraded image fusion.

**Competing Methods.** To comprehensively evaluate the performance of our proposed framework, we conducted experiments on three typical image restoration tasks: image dehazing, denoising, and deblurring. We compared our method with three state-of-the-art models that jointly address image restoration and fusion: AWFusion (Li et al., 2024b), DRMF (Tang et al., 2024), and Text-IF (Yi et al., 2024). These comparisons were designed to assess the generalization ability of our model in handling multiple types of degradations. In addition, we selected six image fusion models(DenseFuse (Li & Wu, 2018), SwinFuse (Wang et al., 2022b), CDDFuse (Zhao et al., 2023a), SeAFusion (Tang et al., 2022a), MGDN (Guan et al., 2023) and EMMA (Zhao et al., 2024) ) for further evaluation. Since these models lack inherent image restoration capabilities, we first preprocessed the degraded images using the AdaIR model (Cui et al., 2025), an all-in-one image restoration framework, to obtain restored images. The restored images were then fed into the aforementioned fusion models.

## 4.2. Restoration and Fusion Results on DeMMI-RF and EMS Dataset

To illustrate the visual differences in restored images, Fig. 5 shows fusion results of degraded visible-infrared images from our dataset after restoration and fusion. Additional results on the EMS dataset are provided in Appendix A.2. Compared to existing methods (*e.g.*, DenseFuse, CDDFuse, SeAFusion, MGDN, EMMA), our TG-ECNet better preserves color information during fusion. Restoration-fusion unified models like AWFusion, Text-IF and DRMF prioritize fusion over degradation handling, limiting their restoration performance.

To quantitatively evaluate the performance of different methods in restoring the original image quality, we use CC, MSE, PSNR, $N_{abf}$, and MS-SSIM as evaluation metrics, as shown in Table 1. The quantitative performance for each task can be found in the A.4. The quantitative results confirm that

TG-ECNet consistently achieves superior denoising performance and structural preservation.

**Degraded Visible Images with Noise.** Following standard image restoration protocols, we evaluated performance under medium, high, and extreme Gaussian noise conditions. For moderate noise (Fig. 5, first row), TG-ECNet effectively removes noise while preserving structural details, outperforming SeAFusion and EMMA in clarity and information retention. Under high noise (second row), most methods fail to completely eliminate noise, significantly degrading fusion quality. In extreme conditions (third row), while AWFusion and DRMF exhibit excessive blurring and CDDFuse retains noise artifacts, TG-ECNet maintains superior noise suppression and detail preservation, achieving optimal information integration from both modalities.

**Degraded Visible Images with Haze.** In this experiment, we employ a novel atmospheric scattering model that inherently diminishes image brightness and contrast. The degradation model produces images with reduced brightness, lower contrast, and slight color distortions, approximating real-world foggy scenes (see A.1). Infrared images, already limited in brightness and contrast, exacerbate these issues during fusion, leading to severe color deviations in cascade-based two-stage models (*e.g.*, DenseFuse, SwinFuse, CDDFuse). Restoration-fusion unified models often skew toward one task: AWFusion removes haze but alters colors, DRMF prioritizes dehazing over infrared fusion, and Text-IF favors fusion at the cost of haze removal. In contrast, TG-ECNet balances dehazing and fusion, preserving both visible details and infrared information for clearer, more natural results.

**Degraded Visible Images with Defocus Blur.** In the defocus blur scenario, our method demonstrates the ability to capture and restore fine texture details effectively without losing the deblurring effect during fusion. While cascade-based two-stage models provide some degree of restoration, their performance is significantly compromised in the fusion stage due to the inherently low texture clarity of infrared images. This degradation leads to a weakened deblurring effect in the final fusion result, as seen in SeAFusion and EMMA, where noticeable residual blurriness remains. Restoration-fusion unified models also struggle to eliminate blur, though they outperform cascade models. However, their results still fall short compared to TG-ECNet, which maintains a significantly sharper and more detailed fused output, demonstrating its effectiveness in handling defocus blur while preserving structural details.

**Degraded Infrared Images with Stripe Noise.** Stripe noise is a specific degradation type affecting infrared images, and traditional image restoration models perform poorly in removing such noise. Similarly, restoration-fusion unified models tend to prioritize either restoration or fusion. For

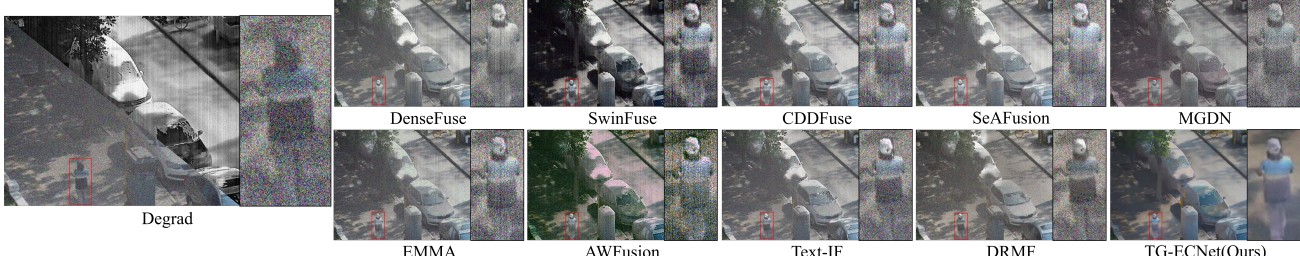

*Figure 6.* Quantitative comparison between related works on multi-degraded scenarios.

*Table 1.* Comparison of average quantitative performance of single tasks across our DeMMI-RF and EMS dataset.

| METHODS | OUR DATASET | | | | | EMS DATASET | | | | |
|---|---|---|---|---|---|---|---|---|---|---|
| | CC | MSE | PSNR | $N_{abf}$ | MS-SSIM | CC | MSE | PSNR | $N_{abf}$ | MS-SSIM |
| DENSEFUSE (LI & WU, 2018) | 0.5185 | 0.0923 | 29.5794 | **0.0863** | 0.2291 | **0.5018** | **0.1090** | 28.9558 | **0.0715** | **0.3313** |
| SWINFUSE (WANG ET AL., 2022B) | 0.5279 | 0.0928 | 29.5190 | 0.1180 | **0.2418** | 0.5002 | 0.1717 | 28.1347 | 0.1176 | 0.1396 |
| CDDFUSE (ZHAO ET AL., 2023A) | 0.5286 | **0.0787** | **29.8441** | 0.1116 | 0.2359 | 0.5005 | 0.1120 | 28.8962 | 0.0933 | 0.3197 |
| SEAFUSION (TANG ET AL., 2022A) | **0.5288** | 0.0904 | 29.5892 | 0.1170 | 0.2265 | **0.5013** | **0.1115** | **28.9057** | 0.0874 | 0.3187 |
| MGDN (GUAN ET AL., 2023) | 0.5279 | **0.0669** | **30.2216** | 0.1195 | 0.2431 | 0.4985 | 0.1314 | 28.7083 | 0.0924 | 0.1763 |
| EMMA (ZHAO ET AL., 2024) | 0.5198 | 0.0821 | 29.7471 | 0.1266 | **0.2384** | 0.4996 | 0.1132 | 28.8699 | 0.0966 | 0.3057 |
| AWFUSION (LI ET AL., 2024B) | 0.5265 | 0.0979 | 29.4554 | 0.1429 | 0.2073 | **0.5013** | 0.2221 | 27.5428 | 0.2796 | 0.1114 |
| TEXT-IF (YI ET AL., 2024) | **0.5309** | 0.0880 | 29.5656 | **0.0804** | 0.2379 | 0.5007 | **0.1115** | 28.9034 | 0.0800 | **0.3210** |
| DRMF (TANG ET AL., 2024) | 0.5249 | 0.0883 | 29.5077 | 0.1203 | 0.2179 | 0.4991 | 0.1261 | 28.6382 | **0.0739** | 0.3158 |
| TG-ECNET(OURS) | **0.5340** | **0.0570** | **30.5822** | **0.0385** | **0.2972** | **0.5035** | **0.0738** | **29.7962** | **0.0405** | **0.4273** |

*Table 2.* Comparison of quantitative performance of multi-tasks and degradation-free task on our DeMMI-RF dataset.

| METHODS | MULTI-TASKS(NOISE+HAZE+DEFOCUSBLUR+STRIPE) | | | | | DEGRADATION-FREE TASK | | | | |
|---|---|---|---|---|---|---|---|---|---|---|
| | CC | MSE | PSNR | $N_{abf}$ | MS-SSIM | CC | MSE | PSNR | $N_{abf}$ | MS-SSIM |
| DENSEFUSE (LI & WU, 2018) | **0.5225** | 0.0950 | 29.2285 | **0.1240** | 0.1740 | 0.5209 | 0.0819 | 29.7344 | **0.0516** | 0.3365 |
| SWINFUSE (WANG ET AL., 2022B) | 0.5075 | 0.0940 | 29.4485 | **0.1010** | **0.2540** | 0.5464 | **0.0659** | **30.2302** | **0.0493** | **0.3882** |
| CDDFUSE (ZHAO ET AL., 2023A) | **0.5225** | 0.0880 | 29.4665 | 0.1330 | **0.1790** | 0.5464 | 0.0722 | 30.0046 | 0.0653 | 0.3480 |
| SEAFUSION (TANG ET AL., 2022A) | **0.5220** | 0.0940 | 29.2615 | **0.1240** | 0.1640 | **0.5471** | 0.0752 | 29.8970 | 0.0645 | 0.3452 |
| MGDN (GUAN ET AL., 2023) | 0.5200 | **0.0790** | **29.6820** | 0.1570 | 0.1750 | **0.5475** | **0.0546** | **30.5682** | 0.0652 | **0.3676** |
| EMMA (ZHAO ET AL., 2024) | 0.5195 | **0.0820** | **29.5760** | 0.1580 | 0.1600 | 0.5452 | 0.0689 | 30.1192 | 0.0682 | 0.3446 |
| AWFUSION (LI ET AL., 2024B) | 0.5190 | 0.0990 | 29.1365 | 0.1970 | 0.1360 | 0.5458 | 0.0730 | 29.9619 | 0.0827 | 0.3244 |
| TEXT-IF (YI ET AL., 2024) | 0.5205 | 0.0970 | 29.1655 | 0.1640 | 0.1620 | 0.5462 | 0.0757 | 29.8425 | 0.0666 | 0.3350 |
| DRMF (TANG ET AL., 2024) | 0.4960 | 0.0900 | 29.3640 | 0.2390 | 0.1560 | 0.5449 | 0.0812 | 29.6828 | 0.0683 | 0.3259 |
| TG-ECNET(OURS) | **0.5245** | **0.0630** | **30.2200** | **0.0110** | **0.2870** | **0.5518** | **0.0455** | **30.9863** | **0.0373** | **0.4300** |

*Table 3.* Ablation studies on our DeMMI-RF dataset.

| SETTING | NOISE AVERAGE $\sigma = 15, 25, 50$ | | | | | STRIPE NOISE | | | | |
|---|---|---|---|---|---|---|---|---|---|---|
| | CC | MSE | PSNR | $N_{abf}$ | MS-SSIM | CC | MSE | PSNR | $N_{abf}$ | MS-SSIM |
| W/O TASK-AWARE GATING | 0.5527 | 0.0430 | 31.136 | 0.028 | 0.411 | 0.5335 | 0.0543 | 30.577 | 0.050 | 0.399 |
| W/O ALL MULTI-EXPERT BLOCKS | **0.5547** | 0.0399 | 31.314 | 0.030 | 0.396 | 0.5306 | 0.0489 | 30.844 | 0.057 | 0.387 |
| W/O MULTI-EXPERT BLOCK IN RESTORATION STAGE | 0.5511 | 0.0410 | 30.014 | 0.044 | 0.386 | 0.5106 | 0.0509 | 29.897 | 0.077 | 0.287 |
| W/O MULTI-EXPERT BLOCK IN FUSION STAGE | 0.5523 | 0.0397 | 31.343 | 0.029 | 0.410 | **0.5378** | 0.0484 | 30.894 | 0.051 | 0.391 |
| W/O TWO-STAGE TRAINING STRATEGY | 0.5537 | 0.0451 | 31.010 | 0.033 | 0.394 | 0.5313 | 0.0595 | 30.310 | 0.129 | 0.278 |
| OURS | 0.5536 | **0.0386** | **31.388** | **0.026** | **0.414** | 0.5310 | **0.0472** | **30.932** | **0.048** | **0.412** |

example, DRMF focuses on removing stripe noise but simultaneously eliminates a substantial amount of useful infrared details, leading to an unnatural fusion result. Conversely, Text-IF prioritizes image fusion, resulting in incomplete noise removal. TG-ECNet achieves a better trade-off, effectively suppressing stripe noise while preserving the fine details of the infrared image, leading to a cleaner and more informative fusion result.

From these experimental settings, it is evident that TG-ECNet consistently outperforms other methods by maintaining a stable restoration effect across different degradation

types. Existing cascaded methods that first restore and then fuse images often perform poorly, leading to noise enhancement or loss of critical image information. Unlike existing methods, TG-ECNet effectively balances these two tasks, demonstrating strong adaptability to complex degradation conditions and ensuring superior fusion quality.

### 4.3. Ablation Study

In this experiment, we investigate the effect of different components on task-specific restoration and fusion performance

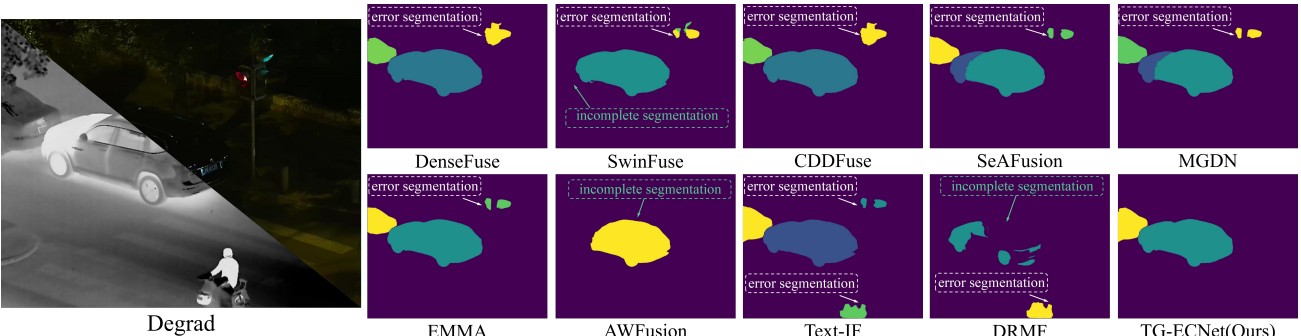

*Figure 7.* Visual results in segmentation scenario when the target is "car" with Grounded-SAM.

*Table 4.* Object Detection Evaluation on Noise($\sigma = 50$).

| METRIC | DENSEFUSE | SWINFUSE | CDDFUSE | SEAFUSION | MGDN | EMMA | AWFUSION | TEXT-IF | DRMF | OURS |
|---|---|---|---|---|---|---|---|---|---|---|
| MAP50 | 0.926 | 0.914 | 0.935 | 0.959 | 0.901 | 0.963 | 0.921 | 0.963 | 0.95 | **0.969** |
| AP(0.5:0.95) | 0.526 | 0.476 | 0.511 | 0.52 | 0.488 | 0.523 | 0.519 | 0.536 | 0.493 | **0.537** |

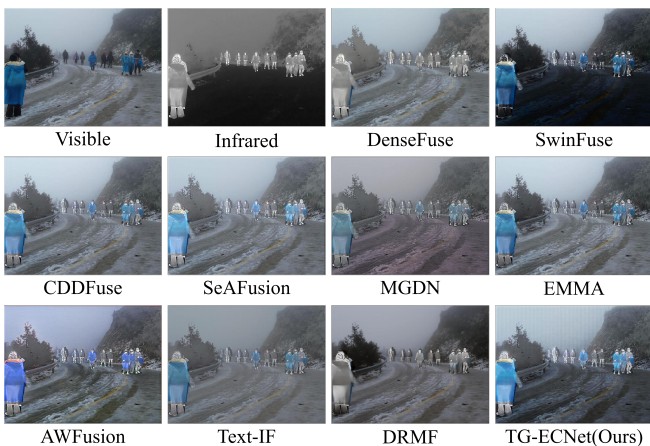

*Figure 8.* Visual results on real-world haze images.

using a series of ablation studies. The results are summarized in Table 3, which shows the performance metrics for noise average ($\sigma = 15, 25, 50$) and stripe noise scenarios.

**w/o Task-aware Gating.** The first ablation study examines the effect of removing the task-aware gating, which is used to guide the task-specific restoration process and multi-modal image fusion process. In this configuration, the model operates without task-aware gating, which may lead to suboptimal task-specific guidance during the fusion and restoration phases. As shown in the Table 3, the model performs slightly worse in terms of CC, MSE, PSNR, and MS-SSIM metrics, as the task prompt plays a significant role in guiding the network toward optimal restoration for each specific task.

**w/o Multi-Expert Block.** We conducted three comparative experiments without expert blocks, namely removing the restoration expert block, fusion expert block, and removing both, which is responsible for expert selection and task adaptation. By doing so, the model loses its ability to selectively choose the most appropriate expert for each task, potentially leading to performance degradation. The results show that removing the multi-expert block has a slight negative effect

on performance, with a reduction in PSNR and MS-SSIM, suggesting that the multi-expert block plays a critical role in guiding the model towards the most effective expert for each task. The drop in $N_{abf}$ further supports the importance of selection in fine-tuning restoration and fusion. Among these three settings, it can be seen that restoration expert blocks are more important for overall performance, while fusion slightly optimizes performance.

**w/o Two-stage Training Strategy.** Finally, we examine the effect of the two-stage training strategy by comparing it with a single-stage network. In this experiment, the two-stage training strategy involves first training the restoration network and then freezing part of its parameters while using the fusion module to generate the fused restoration results. This staged approach ensures a better balance between restoration and fusion tasks. The results show that the two-stage training strategy performs slightly better than the single-stage network in terms of CC, PSNR, and MS-SSIM, confirming that freezing certain parameters during the fusion stage helps improve the fusion quality. While the difference in $N_{abf}$ is minimal, the two-stage strategy proves to be a more effective approach for achieving task-specific restoration and fusion.

### 4.4. Discussion

**Multiple Degradation Scenario All-in-One Image Fusion.** In this experiment, we extend our evaluation to more complex scenarios by combining multiple degradation types for restoration and fusion. In addition to single-degradation tasks, we test the model's performance on mixed degradation tasks, comparing our restoration-fusion unified model with other methods. The experiment covers a total of 9 different degradation combinations, which are derived from the same scenarios. We have also set 8 other scenarios, which can be found in our DeMMI-RF dataset. However, only the most challenging settings are presented in Table 2, with the remaining combinations provided in the A.6. As shown

in Table 2, TG-ECNet outperforms all of other methods. For the Noise50 + Defocusblur + Haze + Stripe condition, TG-ECNet achieves higher PSNR, Nabf, and MS-SSIM, indicating superior performance in both restoration and fusion. In Fig. 6, which represents the visualization results, most methods fail to eliminate all of the degradations, resulting in inferior image quality of the fusion results. Notably, Text-IF suppressed blurring and haze, but it cannot simultaneously removing noise. DenseFuse suffers from a significant loss of visible spectrum information. In contrast, TG-ECNet demonstrates superior degradation suppression while effectively preserving fine details, ensuring that both infrared and visible information are meaningfully integrated.

**Degradation-Free Scenario Image Fusion.** In order to verify the excellent fusion effect of the experiment on conventional images, we selected some data without degradation and directly compared it with all image fusion algorithms. The experimental results are shown in Table 2. Based on our experimental framework, we employ a two-stage evaluation process to assess comparative fusion algorithms. This methodology requires images to undergo restoration through the preprocessing network prior to fusion, introducing a cascaded processing pipeline. However, this sequential architecture inevitably leads to incremental information degradation, as quantitatively evidenced by the performance metrics. Restoration-fusion unified models did not take this situation into consideration, obtaining a relatively weak performance. TG-ECNet has a moderate increase in all metrics.

**Real-World Degradation Scenario Image Fusion.** To verify the effectiveness of the model on real-world data, we used real collected data AWMM (Li et al., 2024b) for testing, and the qualitative results are shown in Fig. 8. The results of TG-ECNet effectively suppress haze to make the image clearer, while fusing infrared information simultaneously, and the overall effect is very close to that of data provider AWFusion.

### 4.5. Detection and Segmentation Evaluation

**Detection Evaluation.** We fed the experimental results generated by ten models into the YOLOv5 model, along with degraded input images and clean images. The dataset was split into a 7:3 training-to-testing ratio, with 50 training epochs and an image resolution of $640 \times 640$. The detection metrics are presented in Table 4. Our method achieves state-of-the-art performance in terms of mean Average Precision (mAP) and AP(0.5:0.95), outperforming all compared methods. This demonstrates the effectiveness of our framework in accurately detecting and localizing objects under various degradation conditions. The superior performance in these metrics highlights the robustness and generalization ability of our approach.

**Segmentation Evaluation.** We fed the experimental re-

sults generated by ten models into the Grounded-SAM model (Ren et al., 2024), using the pretrained model parameters of Grounded-SAM and selecting "car" as the prompt for the image shown in Fig. 7 to obtain the segmentation results. As shown in Fig. 7, almost all of the other methods mistakenly divide the traffic light or the electric bicycle into cars except AWFusion. Besides, SwinFuse, AwFusion, and DRMF cannot segment two cars. However, we successfully segmented the clear outlines of the two cars and did not mix them together.

## 5. Conclusion

We propose the Task-Gated Multi-Expert Collaboration Network (TG-ECNet), a novel framework for degraded multimodal image fusion. TG-ECNet unifies image restoration and fusion into a single end-to-end model, addressing challenges posed by combined degradations like noise, blur, haze, and stripe noise. The core innovation is its task-aware gating, which integrates degradation-aware gating in the encoder and fusion-aware gating in the decoder to adapt to diverse degradation types. A multi-expert collaborative framework and two-stage training strategy ensure balanced optimization. Experiments on benchmark datasets show that TG-ECNet outperforms state-of-the-art methods, improving fusion quality and robustness in challenging environments.

## Acknowledgements

This work was supported in part by the National Science and Technology Major Project under Grant 2022ZD0116500, in part by the National Natural Science Foundation of China under Grants 62436002, 62222608, and U23B2049, in part by the Postdoctoral Fellowship Program of CPSF under Grant Number GZB20250395, in part by Tianjin Natural Science Funds for Distinguished Young Scholar under Grant 23JCJQJC00270, in part by the Fundamental Research Funds for the Central Universities under Grant 3208002502C2, in part by Zhejiang Provincial Natural Science Foundation of China under Grant LD24F020004, and in part by the Huawei Ascend Computing Platform.

## Impact Statement

This paper propose a unified framework for degraded multimodal image restoration and fusion, which bridges different tasks together through a two-stage training strategy to learn inter-task information while avoiding mutual interference, enabling all-in-one processing. This work construct a large-scale degraded multi-modal image fusion benchmark, DeMMI-RF, which contains more than $30,000$ multi-modal data of different degradation types, including those from UAVs and driving viewpoints. This dataset will be of greater significance to the field.

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

# A. Appendix.

## A.1. The haze Setting

The traditional method for adding haze to an image is the atmospheric scattering model, with the formula:

$$J(x) = \frac{1}{t(x)}I(x) - A\frac{1}{t(x)} + b \tag{7}$$

Where $t(x)$ indicates image with haze, $J(x)$ indicates image without haze, $A$ indicates atmospheric light intensity. In our work, we set $A$ as:

$$\overline{A} = MEAN(J(x)) \tag{8}$$

which avoids the image being too bright and makes it close to reality.

## A.2. Qualitative performance on each task in EMS dataset.

As shown in Fig. 9, qualitative comparison has been made on each task in EMS dataset. Since the EMS dataset utilized some lower clarity datasets, the overall effect appears blurry.

**Degraded Visible Images with Noise.** In the denoising task, our method effectively suppresses noise and makes the edge information of the image clearer. Except for Text-IF, most methods cannot effectively suppress noise. Although Text-IF can better preserve color information, it does not preserve contour information very well. This is enough to demonstrate the effectiveness of our method.

**Degraded Visible Images with Haze.** The performance of different methods in dehazing tasks varies. DenseFuse, CDDFuse, SwinFuse, EMMA, and DRMF cannot completely remove haze. SwinFuse, MGDN, and AWFusion have caused significant color distortion. Although Text-IF removed most of the haze, it did not completely preserve the profile of people and car, and our method outperformed it in this regard.

**Degraded Visible Images with Defocusblur.** MGDN and DRMF cannot fully restore clear images in defocusbluring tasks. The remaining methods have achieved the effect of deblurring, but the image information is too dependent on infrared images, resulting in the loss of color details.

**Degraded Infrared Images with Stripe Noise.** In the stripe task, our method not only effectively suppresses stripe noise, but also achieves a balance between color information and thermal information.

In summary, our method still achieved excellent results on the EMS dataset, which proves the robustness of our method.

*Table 5.* Mixture of experts hyperparameter selection.

| SETTING | ALL TASK AVERAGE | | | | |
|---|---|---|---|---|---|
| | CC | MSE | PSNR | $N_{abf}$ | MS-SSIM |
| 2 IN 3 | **0.5531** | 0.0408 | 31.2374 | 0.0258 | 0.4087 |
| 3 IN 5 | 0.5529 | 0.0411 | 31.2207 | 0.0269 | 0.4056 |
| 4 IN 7 | 0.5525 | 0.0446 | 31.0421 | 0.0233 | 0.4053 |
| 5 IN 9 | 0.5520 | 0.0409 | 31.2360 | 0.0247 | 0.4069 |
| 6 IN 11 | 0.5512 | **0.0397** | **31.3086** | **0.0223** | **0.4206** |
| 7 IN 13 | 0.5498 | 0.0408 | 31.2459 | 0.0235 | 0.4113 |
| 3 IN 11 | 0.5512 | 0.0420 | 31.1842 | 0.0270 | 0.4074 |
| 9 IN 11 | 0.5519 | 0.0411 | 31.2394 | 0.0226 | 0.4094 |
| 11 IN 11 | 0.5523 | 0.0410 | 31.2273 | 0.0232 | 0.4115 |

*Table 6.* Mixture of experts block setting.

| SETTING | ALL TASK AVERAGE | | | | |
|---|---|---|---|---|---|
| | CC | MSE | PSNR | $N_{abf}$ | MS-SSIM |
| W/O MoE-2&3(EX1) | 0.5519 | 0.0403 | 31.2981 | 0.0224 | 0.4180 |
| W/O MoE-1&3(EX2) | 0.5517 | 0.0440 | 31.0731 | 0.0302 | 0.4037 |
| W/O MoE-1&2(EX3) | 0.5525 | **0.0397** | 31.2858 | 0.0232 | 0.4165 |
| W/O MoE-1(EX23) | 0.5530 | 0.0423 | 31.1435 | 0.0264 | 0.4123 |
| W/O MoE-2(EX13) | 0.5516 | 0.0403 | 31.2642 | 0.0217 | 0.4143 |
| W/O MoE-3(EX12) | 0.5531 | 0.0414 | 31.2007 | 0.0279 | 0.4137 |
| W/O FUSIONMOE | 0.5514 | 0.0425 | 31.1531 | 0.0271 | 0.4144 |
| ALL EXPERT | **0.5537** | **0.0397** | **31.3086** | **0.0216** | **0.4206** |

## A.3. Mixture of Experts block Setting

In our experiments, we explored the optimal configuration for the Mixture of Experts (MoE) system in a subset of the DeMMI-RF dataset by testing various expert selection modes, including no experts, 3 experts selecting 2, 5 experts selecting 3, 7 experts selecting 4, 9 experts selecting 5, 11 experts selecting 6, and 13 experts selecting 7. The results, as shown in Table 5, indicate that the 11 experts selecting 6 configuration achieves the best performance, highlighting the importance of balancing task specialization and computational efficiency. Additionally, we compared different expert quantities within the 11-expert system (11 selecting 3, 11 selecting 6, and 11 selecting 9) and found that increasing the number of selected experts does not always improve restoration or fusion performance, further confirming the need for an optimal expert selection strategy.

We integrated Multi-Expert (MoE) collaborative modules before the transformer blocks of the first three encoders and after the last decoder's transformer block. Ablation studies confirm their effectiveness: in the restoration module, a three-level MoE system outperformed partial configurations (one or two levels), with full retention yielding optimal feature extraction (Table 6). Similarly, removing the MoE from the fusion module severely degraded performance, highlighting its necessity for feature aggregation and high-quality fusion.

Overall, these experiments validate the effectiveness of our MoE-based design, particularly the 11 experts selecting 6 configuration and the strategic placement of MoE systems across the restoration and fusion modules. These findings

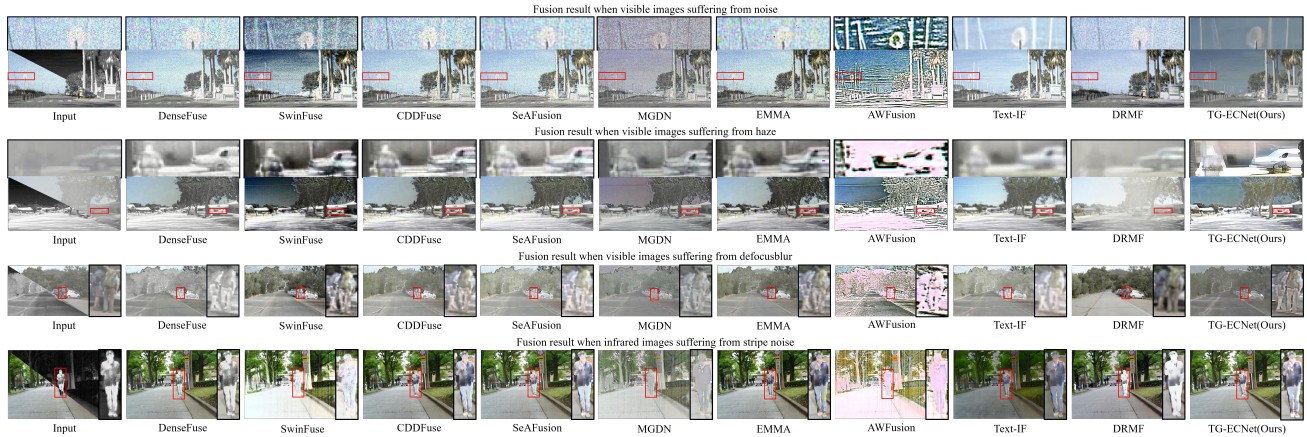

*Figure 9.* Qualitative comparisons of various methods on EMS dataset.

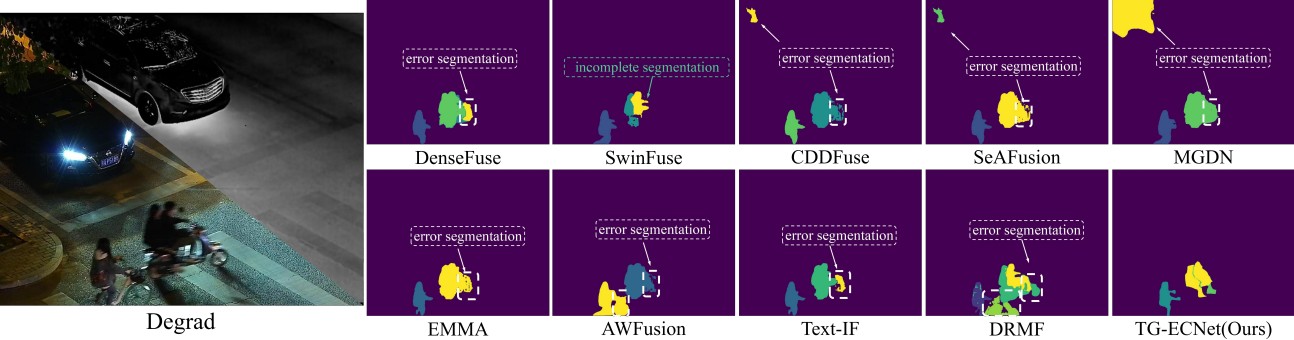

*Figure 10.* Visual results in segmentation scenario when the target is person with Grounded-SAM.

not only optimize the performance of our framework but also provide valuable insights for future research on task-specific expert systems in multi-modal image processing.

### A.4. Quantitative performance on each task

**Degraded Visible Images with Noise.** As shown in Table 7 and Table 8, we achieved the best results in multiple metrics for both settings. DenseFuse relies more on infrared images, which, to some extent, suppresses Gaussian noise in visible light images, while Text IF has better noise recognition ability and achieves relatively good results. Other methods, such as CDDFuse, MGDN, and AWFusion only perform well on one or two metrics.

**Degraded Visible Images with Haze in DeMMI-RF and EMS dataset.** As shown in Table 9, in the dehazing task, due to the different methods of generating haze, the performance between the two datasets is different. Our method achieved the best results in multiple metrics on both datasets. On the EMS dataset, DRMF achieved a suboptimal level, while AWFusion, which specializes in image dehazing, achieved the best performance on CC.

**Degraded Visible Images with Defocusblur in DeMMI-RF and EMS dataset.** As shown in Table 10, our method

achieved multiple optimal metrics on two datasets in the deblurring task. Due to MGDN's ability to achieve multi focus image fusion, it has achieved a suboptimal level.

**Degraded infrared Images with Stripe Noise in DeMMI-RF and EMS dataset.** As shown in Table 11, our method achieved the best performance on all metrics in the task of removing stripe noise on two datasets.

Our experimental results demonstrate that the proposed method achieves superior and robust performance across both benchmark datasets. Quantitatively, the approach shows significant improvements in key metrics, outperforming current state-of-the-art methods. The method's strength lies in its ability to maintain an optimal balance between noise suppression and detail preservation.

### A.5. Other Segmentation Results

We also fed the experimental results, with three persons riding a bicycle or an electric bicycle, generated by ten models into the Grounded-SAM model (Ren et al., 2024), using the pretrained model parameters of Grounded-SAM and selecting "person" as the prompt for the image shown in Fig. 10 to obtain the segmentation results. As shown in Fig. 10, CDDFuse, SeAFusion, and MGDN mistook the

*Table 7.* Quantitative comparison on our DeMMI-RF dataset in different Noise settings.

| METHODS | DENOISE | | | | | | | | | |
| --- | --- | --- | --- | --- | --- | --- | --- | --- | --- | --- |
| | OUR DATASET (NOISE$_{\sigma=15}$) | | | | | OUR DATASET (NOISE$_{\sigma=25}$) | | | | |
| | CC | MSE | PSNR | $N_{abf}$ | MS-SSIM | CC | MSE | PSNR | $N_{abf}$ | MS-SSIM |
| DENSEFUSE (LI & WU, 2018) | 0.5200 | 0.1028 | 29.2159 | 0.0850 | 0.2139 | 0.5198 | 0.0927 | 29.4729 | 0.0805 | 0.2228 |
| SWINFUSE (WANG ET AL., 2022B) | 0.5272 | 0.0977 | 29.2750 | 0.1453 | 0.1930 | 0.5372 | 0.0936 | 29.3787 | 0.1377 | 0.2116 |
| CDDFUSE (ZHAO ET AL., 2023A) | 0.5266 | 0.0896 | 29.4820 | 0.1265 | 0.2135 | 0.5366 | 0.0798 | 29.7036 | 0.1201 | 0.2194 |
| SEAFUSION (TANG ET AL., 2022A) | 0.5269 | 0.1062 | 29.1165 | 0.1303 | 0.2075 | 0.5362 | 0.0926 | 29.4142 | 0.1209 | 0.2170 |
| MGDN (GUAN ET AL., 2023) | 0.5262 | 0.0746 | 29.8921 | 0.1299 | 0.2286 | 0.5350 | 0.0656 | 30.1583 | 0.1268 | 0.2291 |
| EMMA (ZHAO ET AL., 2024) | 0.5190 | 0.0902 | 29.4201 | 0.1758 | 0.1961 | 0.5280 | 0.0795 | 29.7504 | 0.1728 | 0.1917 |
| AWFUSION (LI ET AL., 2024B) | 0.5244 | 0.1108 | 29.0827 | 0.1501 | 0.1874 | 0.5340 | 0.0890 | 29.5590 | 0.1399 | 0.1961 |
| TEXT-IF (YI ET AL., 2024) | 0.5253 | 0.0957 | 29.3726 | 0.0773 | 0.2314 | 0.5345 | 0.0858 | 29.6254 | 0.0740 | 0.2385 |
| DRMF (TANG ET AL., 2024) | 0.5211 | 0.1062 | 29.1004 | 0.1702 | 0.1803 | 0.5308 | 0.0835 | 29.6678 | 0.1675 | 0.1882 |
| TG-ECNET(OURS) | 0.5274 | 0.0602 | 30.3239 | 0.0395 | 0.2877 | 0.5382 | 0.0570 | 30.4800 | 0.0378 | 0.2858 |

*Table 8.* Quantitative comparison on our DeMMI-RF and EMS dataset in Noise Setting.

| METHODS | DENOISE | | | | | | | | | |
| --- | --- | --- | --- | --- | --- | --- | --- | --- | --- | --- |
| | OUR DATASET (NOISE$_{\sigma=50}$) | | | | | EMS DATASET | | | | |
| | CC | MSE | PSNR | $N_{abf}$ | MS-SSIM | CC | MSE | PSNR | $N_{abf}$ | MS-SSIM |
| DENSEFUSE (LI & WU, 2018) | 0.5133 | 0.0890 | 29.5150 | 0.1026 | 0.2099 | 0.4760 | 0.1040 | 29.0510 | 0.1080 | 0.1720 |
| SWINFUSE (WANG ET AL., 2022B) | 0.5232 | 0.0922 | 29.3806 | 0.2026 | 0.1847 | 0.4755 | 0.1360 | 28.4600 | 0.1970 | 0.0720 |
| CDDFUSE (ZHAO ET AL., 2023A) | 0.5245 | 0.0785 | 29.7360 | 0.1751 | 0.1968 | 0.4730 | 0.1150 | 28.8275 | 0.1820 | 0.1540 |
| SEAFUSION (TANG ET AL., 2022A) | 0.5257 | 0.0846 | 29.5466 | 0.1833 | 0.1840 | 0.4745 | 0.1060 | 28.9915 | 0.1330 | 0.1620 |
| MGDN (GUAN ET AL., 2023) | 0.5225 | 0.0672 | 30.0439 | 0.2010 | 0.1885 | 0.4755 | 0.0887 | 29.4320 | 0.1450 | 0.0950 |
| EMMA (ZHAO ET AL., 2024) | 0.5152 | 0.0794 | 29.6692 | 0.2538 | 0.1518 | 0.4735 | 0.1140 | 28.8270 | 0.1690 | 0.1520 |
| AWFUSION (LI ET AL., 2024B) | 0.5241 | 0.0810 | 29.6963 | 0.1843 | 0.1735 | 0.4890 | 0.1630 | 28.0385 | 0.3980 | 0.0450 |
| TEXT-IF (YI ET AL., 2024) | 0.5251 | 0.0785 | 29.7154 | 0.1263 | 0.1934 | 0.4765 | 0.1070 | 28.9845 | 0.0930 | 0.1860 |
| DRMF (TANG ET AL., 2024) | 0.5162 | 0.0880 | 29.4381 | 0.2567 | 0.1275 | 0.4695 | 0.1330 | 28.4930 | 0.1330 | 0.1680 |
| TG-ECNET(OURS) | 0.5281 | 0.0546 | 30.5374 | 0.0374 | 0.2678 | 0.4735 | 0.0880 | 29.4120 | 0.0470 | 0.2300 |

ground as a person. Besides, almost all of the methods segmented bicycles as part of person and could not separate two people. However, we successfully segmented the clear outlines of the three people and did not mix them together.

## A.6. Multiple Degradation Results

As shown in Table 12, 13, 14, and 15, TG-ECNet outperforms both DRMF and Text-IF across diverse degradation conditions shown. These results further demonstrate that TG-ECNet consistently outperforms existing methods, even under more complex degradation scenarios.

## A.7. Weight Setting Verifications

In the experiment, after obtaining the restored visible and infrared features, we use trainable weights to obtain fused features. To verify the effectiveness of this experimental setting, we conducted the following comparative experiments: a).visible modality dominant(VIS Major), b).infrared modality dominant(IR Major), c).direct VIS-IR modality addition(plus). Among them, major means to obtain fused features with a ratio of 90 percent and another modality of 10 percent. As shown in Table 16, our setting can achieve better performance.

## A.8. Computing Efficiency Verifications

As shown in the Table 17, we compared the computational efficiency of different methods and different MoE settings, which can demonstrate the effectiveness of our setting.

*Table 9.* Quantitative comparison on our DeMMI-RF and EMS dataset in Haze Setting.

| METHODS | DEHAZING | | | | | | | | | |
|---|---|---|---|---|---|---|---|---|---|---|
| | OUR DATASET | | | | | EMS DATASET | | | | |
| | CC | MSE | PSNR | $N_{abf}$ | MS-SSIM | CC | MSE | PSNR | $N_{abf}$ | MS-SSIM |
| DENSEFUSE (LI & WU, 2018) | 0.5205 | 0.0977 | 29.4179 | 0.0705 | 0.2418 | 0.4795 | 0.0900 | 29.3665 | 0.0603 | 0.1998 |
| SWINFUSE (WANG ET AL., 2022B) | 0.5324 | 0.1067 | 29.0076 | 0.0523 | 0.3128 | 0.4835 | 0.1430 | 28.3600 | 0.0950 | 0.0950 |
| CDDFUSE (ZHAO ET AL., 2023A) | 0.5361 | 0.0761 | 29.8585 | 0.0607 | 0.2829 | 0.4785 | 0.1000 | 29.1615 | 0.0797 | 0.2111 |
| SEAFUSION (TANG ET AL., 2022A) | 0.5367 | 0.0917 | 29.5512 | 0.0788 | 0.2574 | 0.4790 | 0.0960 | 29.2265 | 0.0831 | 0.1978 |
| MGDN (GUAN ET AL., 2023) | 0.5372 | 0.0706 | 30.0563 | 0.0638 | 0.2761 | 0.4810 | 0.0840 | 29.5095 | 0.0680 | 0.1210 |
| EMMA (ZHAO ET AL., 2024) | 0.5187 | 0.0933 | 29.3895 | 0.0476 | 0.3383 | 0.4775 | 0.1030 | 29.0635 | 0.0877 | 0.2000 |
| AWFUSION (LI ET AL., 2024B) | 0.5342 | 0.1101 | 29.1808 | 0.1043 | 0.2394 | 0.4930 | 0.1720 | 27.9195 | 0.3630 | 0.0550 |
| TEXT-IF (YI ET AL., 2024) | 0.5375 | 0.0947 | 29.4805 | 0.0732 | 0.2561 | 0.4785 | 0.1030 | 29.0580 | 0.0855 | 0.1927 |
| DRMF (TANG ET AL., 2024) | 0.5239 | 0.0796 | 29.7185 | 0.0571 | 0.2969 | 0.4865 | 0.0940 | 29.2695 | 0.0253 | 0.2725 |
| TG-ECNET(OURS) | 0.5394 | 0.0621 | 30.3676 | 0.0419 | 0.3204 | 0.4855 | 0.0720 | 29.8560 | 0.0242 | 0.2732 |

*Table 10.* Quantitative comparison on our DeMMI-RF and EMS dataset in Blur Setting.

| METHODS | DEBLUR | | | | | | | | | |
|---|---|---|---|---|---|---|---|---|---|---|
| | OUR DATASET | | | | | EMS DATASET | | | | |
| | CC | MSE | PSNR | $N_{abf}$ | MS-SSIM | CC | MSE | PSNR | $N_{abf}$ | MS-SSIM |
| DENSEFUSE (LI & WU, 2018) | 0.5169 | 0.0851 | 30.1249 | 0.0507 | 0.2772 | 0.4750 | 0.1040 | 29.0505 | 0.0650 | 0.1920 |
| SWINFUSE (WANG ET AL., 2022B) | 0.5236 | 0.0804 | 30.3363 | 0.0436 | 0.3191 | 0.4745 | 0.1320 | 28.5435 | 0.1040 | 0.0860 |
| CDDFUSE (ZHAO ET AL., 2023A) | 0.5253 | 0.0745 | 30.2695 | 0.0557 | 0.2880 | 0.4720 | 0.1170 | 28.8010 | 0.0850 | 0.1820 |
| SEAFUSION (TANG ET AL., 2022A) | 0.5250 | 0.0842 | 30.0952 | 0.0598 | 0.2843 | 0.4735 | 0.1100 | 28.9290 | 0.0850 | 0.1800 |
| MGDN (GUAN ET AL., 2023) | 0.5251 | 0.0599 | 30.8097 | 0.0541 | 0.3119 | 0.4755 | 0.0873 | 29.2546 | 0.0740 | 0.1130 |
| EMMA (ZHAO ET AL., 2024) | 0.5214 | 0.0742 | 30.2619 | 0.0470 | 0.2858 | 0.4720 | 0.1190 | 28.7475 | 0.0850 | 0.1860 |
| AWFUSION (LI ET AL., 2024B) | 0.5226 | 0.0886 | 29.9665 | 0.0847 | 0.2753 | 0.4860 | 0.1550 | 28.1520 | 0.3350 | 0.0590 |
| TEXT-IF (YI ET AL., 2024) | 0.5371 | 0.0915 | 29.4744 | 0.0565 | 0.2686 | 0.4760 | 0.1060 | 29.0125 | 0.0900 | 0.1860 |
| DRMF (TANG ET AL., 2024) | 0.5342 | 0.0968 | 29.2727 | 0.0338 | 0.2576 | 0.4680 | 0.1310 | 28.5305 | 0.0630 | 0.1940 |
| TG-ECNET(OURS) | 0.5409 | 0.0525 | 31.0968 | 0.0274 | 0.3275 | 0.4775 | 0.0850 | 29.4790 | 0.0360 | 0.2500 |

*Table 11.* Quantitative comparison on our DeMMI-RF and EMS dataset in Stripe Setting.

| METHODS | DESTRIPE | | | | | | | | | |
|---|---|---|---|---|---|---|---|---|---|---|
| | OUR DATASET | | | | | EMS DATASET | | | | |
| | CC | MSE | PSNR | $N_{abf}$ | MS-SSIM | CC | MSE | PSNR | $N_{abf}$ | MS-SSIM |
| DENSEFUSE (LI & WU, 2018) | 0.5210 | 0.0893 | 29.4865 | 0.1503 | 0.1829 | 0.5200 | 0.1160 | 28.8095 | 0.0670 | 0.4360 |
| SWINFUSE (WANG ET AL., 2022B) | 0.5250 | 0.0904 | 29.4007 | 0.1710 | 0.1826 | 0.5165 | 0.1970 | 27.9010 | 0.1070 | 0.1800 |
| CDDFUSE (ZHAO ET AL., 2023A) | 0.5231 | 0.0759 | 29.8123 | 0.1661 | 0.1824 | 0.5195 | 0.1130 | 28.8715 | 0.0770 | 0.4200 |
| SEAFUSION (TANG ET AL., 2022A) | 0.5226 | 0.0857 | 29.5779 | 0.1624 | 0.1757 | 0.5200 | 0.1170 | 28.8010 | 0.0780 | 0.4200 |
| MGDN (GUAN ET AL., 2023) | 0.5216 | 0.0663 | 30.1047 | 0.1812 | 0.1852 | 0.5140 | 0.1640 | 28.2040 | 0.0900 | 0.2250 |
| EMMA (ZHAO ET AL., 2024) | 0.5154 | 0.0780 | 29.7903 | 0.1068 | 0.2336 | 0.5180 | 0.1140 | 28.8630 | 0.0840 | 0.3980 |
| AWFUSION (LI ET AL., 2024B) | 0.5204 | 0.1118 | 29.0121 | 0.2304 | 0.1329 | 0.5100 | 0.2650 | 27.1820 | 0.2170 | 0.1540 |
| TEXT-IF (YI ET AL., 2024) | 0.5216 | 0.0790 | 29.7895 | 0.0876 | 0.2224 | 0.5180 | 0.1160 | 28.8195 | 0.0730 | 0.4180 |
| DRMF (TANG ET AL., 2024) | 0.5183 | 0.0717 | 29.9618 | 0.0814 | 0.2305 | 0.5170 | 0.1310 | 28.5460 | 0.0740 | 0.3920 |
| TG-ECNET(OURS) | 0.5259 | 0.0572 | 30.4625 | 0.0525 | 0.2762 | 0.5215 | 0.0680 | 29.9525 | 0.0440 | 0.5560 |

*Table 12.* Multiple degradation all-in-one results in Appendix A.6.

| METHODS | DEFOCUS+HAZE | | | | | NOISE50+HAZE+STRIPE | | | | |
|---|---|---|---|---|---|---|---|---|---|---|
| | CC | MSE | PSNR | $N_{abf}$ | MS-SSIM | CC | MSE | PSNR | $N_{abf}$ | MS-SSIM |
| DRMF (TANG ET AL., 2024) | 0.5025 | 0.0770 | 29.851 | **0.008** | 0.310 | 0.4945 | 0.0720 | 29.788 | 0.277 | 0.171 |
| TEXT-IF (YI ET AL., 2024) | **0.5250** | 0.0930 | 29.272 | 0.026 | 0.278 | 0.5620 | 0.0900 | 29.295 | 0.170 | 0.160 |
| OURS | 0.5225 | **0.0610** | **30.298** | 0.017 | **0.315** | **0.5650** | **0.0460** | **30.733** | **0.022** | **0.273** |

*Table 13.* Multiple degradation all-in-one results in Appendix A.6.

| METHODS | NOISE15+DEFOCUS | | | | | NOISE15+HAZE | | | | |
|---|---|---|---|---|---|---|---|---|---|---|
| | CC | MSE | PSNR | $N_{abf}$ | MS-SSIM | CC | MSE | PSNR | $N_{abf}$ | MS-SSIM |
| DRMF (TANG ET AL., 2024) | **0.5260** | 0.1490 | 28.202 | 0.086 | 0.244 | **0.4335** | 0.0840 | 29.600 | 0.197 | 0.276 |
| TEXT-IF (YI ET AL., 2024) | 0.5240 | 0.1050 | 28.974 | 0.025 | 0.327 | 0.3525 | 0.0930 | 29.239 | 0.038 | 0.383 |
| OURS | 0.5190 | **0.0780** | **29.631** | **0.010** | **0.380** | 0.3435 | **0.0630** | **30.100** | **0.015** | **0.423** |

*Table 14.* Multiple degradation all-in-one results in Appendix A.6.

| METHODS | NOISE25+DEFOCUS | | | | | NOISE25+HAZE | | | | |
|---|---|---|---|---|---|---|---|---|---|---|
| | CC | MSE | PSNR | $N_{abf}$ | MS-SSIM | CC | MSE | PSNR | $N_{abf}$ | MS-SSIM |
| DRMF (TANG ET AL., 2024) | 0.5255 | 0.1470 | 28.232 | 0.150 | 0.208 | 0.4915 | 0.0930 | 29.427 | 0.128 | 0.212 |
| TEXT-IF (YI ET AL., 2024) | **0.5285** | 0.1040 | 28.989 | 0.070 | 0.273 | 0.4985 | 0.0960 | 29.205 | 0.068 | 0.231 |
| OURS | 0.5265 | **0.0720** | **29.776** | **0.012** | **0.370** | **0.4990** | **0.0710** | **29.944** | **0.017** | **0.299** |

*Table 15.* Multiple degradation all-in-one results in Appendix A.6.

| METHODS | NOISE50+DEFOCUS | | | | | NOISE50+HAZE | | | | |
|---|---|---|---|---|---|---|---|---|---|---|
| | CC | MSE | PSNR | $N_{abf}$ | MS-SSIM | CC | MSE | PSNR | $N_{abf}$ | MS-SSIM |
| DRMF (TANG ET AL., 2024) | 0.5035 | 0.1500 | 28.197 | 0.239 | 0.180 | 0.4945 | 0.0700 | 29.847 | 0.239 | 0.178 |
| TEXT-IF (YI ET AL., 2024) | 0.5170 | 0.1120 | 28.822 | 0.177 | 0.201 | **0.5715** | 0.0800 | 29.561 | 0.109 | 0.189 |
| OURS | **0.5200** | **0.0740** | **29.730** | **0.018** | **0.371** | **0.5715** | **0.0500** | **30.559** | **0.035** | **0.267** |

*Table 16.* Weight setting verifications.

| SETTING | ALL TASK AVERAGE | | | | |
|---|---|---|---|---|---|
| | CC | MSE | PSNR | $N_{abf}$ | MS-SSIM |
| VIS MAJOR | 0.5430 | 0.0488 | 29.8896 | 0.0533 | 0.4184 |
| IR MAJOR | 0.5419 | 0.0499 | 29.6783 | 0.0483 | 0.4119 |
| PLUS | **0.5548** | 0.0482 | 29.9257 | 0.0476 | 0.4198 |
| LEARNABLE WEIGHTS | 0.5537 | **0.0397** | **31.3086** | **0.0216** | **0.4206** |

*Table 17.* Computing efficiency verifications.

| METHODS | FPS | TOTAL PARAMS | MOE SETTINGS | FPS | TOTAL PARAMS |
|---|---|---|---|---|---|
| DENSEFUSE (LI & WU, 2018) | 1.64 | 116.001M | 0 | 1.30 | 146.135M |
| SWINFUSE (WANG ET AL., 2022B) | 1.20 | 239.029M | 2IN3 | 1.20 | 150.967M |
| CDDFUSE (ZHAO ET AL., 2023A) | 1.03 | 119.730M | 3IN5 | 1.16 | 153.441M |
| SEAFUSION (TANG ET AL., 2022A) | 1.45 | 164.402M | 4IN7 | 1.12 | 155.915M |
| MGDN (GUAN ET AL., 2023) | 1.39 | 117.910M | 5IN9 | 1.08 | 158.389M |
| EMMA (ZHAO ET AL., 2024) | 1.49 | 121.029M | 6IN11 | 1.03 | 160.863M |
| AWFUSION (LI ET AL., 2024B) | 0.64 | 192.900M | 7IN13 | 0.96 | 167.337M |
| TEXT-IF (YI ET AL., 2024) | 3.09 | 580.850M | 3IN11 | 1.00 | 160.863M |
| DRMF (TANG ET AL., 2024) | 0.20 | 2610.205M | 9IN11 | 0.92 | 160.863M |
| TG-ECNET(OURS) | 1.03 | 160.863M | NOSELECT | 0.91 | 160.863M |

