# OpenReview forum: "Task-Gated Multi-Expert Collaboration Network for Degraded Multi-Modal Image Fusion"
_ICML.cc/2025/Conference — ICML 2025 poster_

### Official Review · Reviewer_5m4E · 2025-03-12

**Overall Recommendation:** 3

**Summary:**

In this paper, authors proposes a framework, TG-ECNet, for degraded multi-modal fusion by unifying restoration and fusion. The key design involves task-gated routing and expert collaboration. The paper conduct a series of experiments to demonstrate the effectiveness of TG-ECNet.

**Claims And Evidence:**

Yes, the claims is supported by evidence.

**Essential References Not Discussed:**

The key references for image restoration are not included in Sec. 2.2, such as Restormer[1], Prompt IR[2], DGUNet[3], etc.

[1] Zamir, Syed Waqas, et al. "Restormer: Efficient transformer for high-resolution image restoration." Proceedings of the IEEE/CVF conference on computer vision and pattern recognition. 2022.

[2] Potlapalli, Vaishnav, et al. "Promptir: Prompting for all-in-one image restoration." Advances in Neural Information Processing Systems 36 (2023): 71275-71293.

[3] Mou, Chong, Qian Wang, and Jian Zhang. "Deep generalized unfolding networks for image restoration." Proceedings of the IEEE/CVF conference on computer vision and pattern recognition. 2022.

**Experimental Designs Or Analyses:**

I have checked the experimental designs. There are some problems and please see more details in the following questions.

**Methods And Evaluation Criteria:**

Yes, it makes sense for the problem.

**Other Comments Or Suggestions:**

Please polish this paper and then seek for possible publication. The current version is somewhat coarse.

**Other Strengths And Weaknesses:**

Strengths:
1. A novel framework to unifiy image restoration and fusion via a task-gated router and multi-expert collaboration.
2. Extensive experiments by authors.
3. The framework demonstrates competitive performance under four degradation scenarios (noise, haze, blur, stripe noise).
4. The paper also conducts experiments on object detection.

---
Weaknesses:
1. This work is still a two-stage method, where the first stage is to restore clean visible images and clean infrared images, and the second stage is to perform image fusion. So why don't I use advanced image restoration algorithms for the first stage? Then, I only need to care about the fusion.
2. How does the "Degradation-Aware Gating" in Figure 2 for the degraded visible and infrared image work? Is this the same as "Task-Gated Router" in Figure 3?
3. According to Figure 2, two U-shaped Transformer network are used for visible and infrared images, respectively. Is this understanding right? The details about this Transformer network are missing.

**Questions For Authors:**

1. There are some errors in Figure 1. From Figure 1, the "Train Stage 1" doesn't involve the loss. But according to Sec. 3.3, the first training stage optimizes the parameters through the restoration loss. Then the fused image ground truth $\boldsymbol{I^C_F}$ is missing in the figure.
2. Again for Figure 1, the symbols are not consistent with the text parts. For example, degraded input $\boldsymbol{I^d_V}$,  $\boldsymbol{I^d_I}$ in Figure 1 and $I^d_V$, $I^d_I$ in Sec. 3.3.
3. For Figure 2, what does "Prompt Components" mean? It is not mentioned in any sections. Then, what do the dotted line and solid line from "Weights" represent?
4. In Sec. 4.1 "Experimental Setting", "the number of experts K and the hyperparameter l balance were heuristically set to 2 and 0.0001 respectively." What is the hyperparameter l, which can not be found in Method. Then, why K is set as 2, where there are at least four degradations involved in the paper.
5. For evaluation metrics, why choosing CC, MSE, PSNR, $N_{abf}$ and MS-SSIM, what do these mean and how they are calculated? In Text-IF and EMMA, they both apply EN, SD, VIF and SCD.

**Relation To Broader Scientific Literature:**

The key contributions of the TG-ECNet framework are connected to advancements in three areas of the broader literature: multi-modal image fusion, degradation-aware restoration, and dynamic network architectures.

**Theoretical Claims:**

There are not theoretical claims in the paper.

---

> ### Author Rebuttal · Authors · 2025-04-01
>
> Thanks for recognizing our contributions in this work. We will reply to your questions in order.
>
> ---
>
> ## Response to the essential references
> #### Concerning the references you suggested, we will incorporate them into our bibliography. Furthermore, our experimental configuration utilizes state-of-the-art image restoration algorithms, AdaIR (ICLR 2025), for preprocessing purposes.
> ---
>
> ## Response to the doubts towards our two-stage method
> #### The traditional two-stage approach (image restoration followed by fusion) may be contradictory. Image restoration tends to eliminate noisy information while image fusion tends to integrate more valid information, but the restoration model may eliminate information of interest for image fusion in the first stage, leading to sub-optimal fusion performance. In our work, the proposed unified framework realizes the divide-and-conquer treatment of different degradation tasks by arranging the corresponding experts to process dynamically through Degradation-Aware Gating performing task routing, while the multi-expert system guided by Fusion-Aware Gating dynamically equilibrates the degree of information retention between the fusion and restoration tasks to realize the better restoration and fusion results.
> #### In comparative evaluations, even when incorporating state-of-the-art network AdaIR (ICLR 2025) for preprocessing and advanced fusion algorithms in cascaded networks ([NewFig3](https://anonymous.4open.science/r/TG-ECNet/NewFig3.png)). Such a setting still underperforms our approach. Our method demonstrates significant advantages through comprehensive experimental validation. Primarily, it effectively minimizes feature loss induced by network cascading while fully exploiting the shared characteristics between image fusion and restoration tasks.
> #### Furthermore, to ensure rigorous validation, we conducted generalization tests using non-degraded images, which consistently showed that dual-network architectures achieve suboptimal performance compared to our method. These results collectively highlight the technical superiority and robustness of our proposed framework.
> ---
>
> ## Response to the "Degradation-Aware Gating" and "Task-Gated Router"
> #### The "Task-Gated Router" in Fig. 3 is an internal component of the "Degradation-Aware Gating" and "Fusion-Aware Gating" in Fig. 2. The Task-Gated Router operates by dynamically integrating input features with task-specific prompt components to generate adaptive routing weights, effectively bridging the gap between task requirements and feature representation.
> #### Meanwhile, the "Degradation-Aware Gating" and "Fusion-Aware Gating" modules work in tandem to perform intelligent expert gating, where the computed weights selectively filter and combine the outputs from different experts. This dual-gating design not only maintains task-relevant feature propagation but also ensures optimal expert utilization based on both degradation characteristics and fusion objectives, thereby enhancing the model's capability to handle complex multi-task scenarios.
> ---
>
> ## Response to the U-shaped Transformer network
> #### The two U-shaped Transformer networks shown in Fig. 2 share identical parameters. During the first-stage training, we developed a base network capable of simultaneously handling degradations in both visible and infrared modalities. It should be noted that such network architecture is quite common in image restoration tasks and does not represent the core innovation of our work. We have added the details of the model architecture in the revised manuscript.
> ---
>
> ## Response to Questions
> #### 1. We apologize for the unclear presentation. First, the loss of Stage 1 is noted in Stage 2 in Fig.2. Second, in the visible-infrared image fusion task, we do not have the ground truth $I_{F}^{C} $.
> #### 2. We thank the reviewers for pointing out the difference in bolding, which we will correct in the revised manuscript.
> #### 3. The components you mentioned are all integral parts of the MoE system. The "Prompt Components" are derived by compressing features into task-specific prompts that encapsulate relevant task information. The ''weight'' is used to combine the outputs from different experts.
> #### 4. Thanks for correcting this typo. In our work, we select the top 6 experts from 11 experts to cope with different degradations.
> #### 5. These five metrics are all classic evaluation indicators in image fusion. Since our method focuses on utilizing expert systems to identify and eliminate degradations rather than directly removing degradations through text prompts and LLMs like Text-IF, it is essential for us to compute PSNR, a metric shared by both image restoration and fusion tasks. This calculation demonstrates the robustness of our experimental results. Also, the other four metrics are commonly adopted in other fusion algorithms as well.
>
> ---
>
> Thanks for your suggestions.

---

> > ### Comment · Reviewer_5m4E · 2025-04-03
> >
> > After rebuttal, my concerns have been addressed. And I recommend acceptance.

---

### Official Review · Reviewer_d8Ku · 2025-03-13

**Overall Recommendation:** 4

**Summary:**

The paper introduces Task-Gated Multi-Expert Collaboration Network (TG-ECNet), a novel framework designed to address the challenges of degraded multimodal image fusion. The key innovation lies in its task-gated router, which integrates degradation-aware gating in the encoder and fusion-aware gating in the decoder to dynamically adapt to various degradation types (e.g., noise, blur, haze, stripe noise) and selectively aggregate features from multiple modalities (e.g., visible and infrared images). The framework also employs a multi-expert collaborative network and a two-stage training strategy to balance restoration and fusion tasks, ensuring high-quality fusion results in complex real-world scenarios. The authors demonstrate the effectiveness of TG-ECNet through extensive experiments on both synthetic and real-world datasets, showing superior performance compared to state-of-the-art methods in terms of restoration and fusion quality.

**Claims And Evidence:**

The claims made in the paper are well-supported by both quantitative and qualitative evidence. The authors provide:

- Quantitative Results: Metrics such as CC (Correlation Coefficient), MSE (Mean Squared Error), PSNR are used to evaluate the performance of TG-ECNet on synthetic and real-world datasets. The results consistently show that TG-ECNet outperforms state-of-the-art methods across various degradation scenarios.

- Qualitative Results: Visual comparisons demonstrate that TG-ECNet effectively removes noise, haze, and stripe noise while preserving fine details and improving fusion quality. The fusion results are clearer and more informative compared to other methods.

- Downstream Task Evaluation: The authors evaluate the impact of TG-ECNet on object detection tasks using YOLOv5, showing that the fused images generated by TG-ECNet lead to higher detection accuracy (mAP and AP(0.5:0.95)) compared to other methods.

However, there are a few areas where the evidence could be strengthened:

- Real-World Generalization: While the synthetic dataset is well-constructed, the evaluation on real-world data is limited. More diverse real-world scenarios (e.g., outdoor scenes, different lighting conditions) should be tested to further validate the robustness of TG-ECNet.

**Essential References Not Discussed:**

The paper covers most of the relevant literature, but there are a few areas where additional references could strengthen the discussion:

- Transformer-Based Fusion: The paper could discuss more recent transformer-based approaches for multimodal image fusion (e.g., SwinFuse, TransFuse) in more detail, as these methods are relevant to the problem of degraded image fusion.

**Experimental Designs Or Analyses:**

The experimental design is sound, with both synthetic and real-world evaluations. The synthetic dataset is well-constructed, and the real-world experiments demonstrate the practical utility of TG-ECNet. However, there are a few areas where the experimental analysis could be improved:

- Real-World Data Diversity: The real-world experiments are limited to a few scenarios. Testing on a wider range of real-world data (e.g., outdoor scenes, different flicker frequencies) would strengthen the claims of generalizability.

- Downstream Task Evaluation: While the paper shows improvements in object detection, it would be beneficial to evaluate the impact of TG-ECNet on other downstream tasks (e.g., segmentation, SLAM) to further demonstrate the versatility of the framework.

**Methods And Evaluation Criteria:**

The proposed methods and evaluation criteria are appropriate for the problem at hand. The use of degradation-aware gating to dynamically adapt to different degradation types and fusion-aware gating to selectively aggregate multimodal features is well-motivated and effectively addresses the ambiguity of degraded image fusion.

**Other Comments Or Suggestions:**

- Expand Ablation Study: The ablation study could be expanded to include more variations of the network architecture to better understand the contribution of each component.

**Other Strengths And Weaknesses:**

- Limited Real-World Evaluation: The real-world experiments are limited in scope, and more diverse scenarios should be tested to validate the generalizability of TG-ECNet.

**Questions For Authors:**

Please refer to the above parts.

**Relation To Broader Scientific Literature:**

The paper is well-situated within the broader literature on multimodal image fusion and degraded image restoration. The authors discuss related works in both conventional video deflickering and event signal filtering, highlighting the unique challenges posed by degraded multimodal images. The proposed method builds on existing ideas (e.g., attention mechanisms, spatio-temporal modeling) but introduces novel components (e.g., task-gated router) to address the specific problem of degraded image fusion.

**Theoretical Claims:**

The paper does not make strong theoretical claims, so there are no theoretical proofs to evaluate. The focus is primarily on the empirical validation of the proposed method.

---

> ### Author Rebuttal · Authors · 2025-04-01
>
> Thanks for recognizing our contributions in this work. We will reply to your questions in order.
>
> ---
>
> ## Response to the real-world experiments
> #### We utilize real-world data from AWMM dataset to validate the robustness of our method in [Newfig2](https://anonymous.4open.science/r/TG-ECNet/NewFig2.png). The dataset consists of images with snow and haze. In our experiment, the results of our method have improved clarity and contrast.
> ---
>
> ## Response to the downstream tasks
> #### In [NewFig6](https://anonymous.4open.science/r/TG-ECNet/NewFig6.png), we employed the unified segmentation network GroundingSAM to evaluate segmentation performance.
> #### For the noisy scenario with $\sigma$=50, we segmented cars in the image, demonstrating that our method produces no false detections.
> #### For infrared images with stripe noise, we segmented humans in the image, showing that our method accurately extracts the contours of all three individuals without missed detections or misclassifying the e-bike as part of the targets.
> ---
>
> ## Response to essential references
> #### We conducted a series of experiments using SwinFuse, with the corresponding results presented in [NewFig1](https://anonymous.4open.science/r/TG-ECNet/NewFig1.jpg), [Newfig2](https://anonymous.4open.science/r/TG-ECNet/NewFig2.png), [NewFig3](https://anonymous.4open.science/r/TG-ECNet/NewFig3.png), [NewFig4](https://anonymous.4open.science/r/TG-ECNet/NewFig4.png), [NewFig5](https://anonymous.4open.science/r/TG-ECNet/NewFig5.png) and [Newfig6](https://anonymous.4open.science/r/TG-ECNet/NewFig6.png). Our method outperformed MGDN across all tasks, and we have also included this literature in our references.
> ---
>
> ## Response to limited ablation studies on critical components
> #### in [NewFig7](https://anonymous.4open.science/r/TG-ECNet/NewFig7.png), we have conducted two other ablation experiments. First, since we use a learnable mask in the fusion stage, we compare this setting with visible modality dominant(VIS Major), infrared modality dominant(IR Major), and direct VIS-IR modality addition(plus). The result demonstrates that our setting is suitable for different scenarios. Second, since we use the multi-expert collaboration module in both stages, we conduct experiments in which only one stage uses the multi-expert collaboration module. In the experiments, we demonstrate that the module is instrumental in both stages.
> ---
> Thanks for your suggestions.
>
> [r1] Li, et al. AWFusion. arXiv:2402.02090,2024.
>
> [r2] Guan, et al. Mutual-guided dynamic network for image fusion. ACM MM, 2023.
>
> [r3] Wang, et al. SwinFuse. IEEE Transactions on Instrumentation and Measurement, 2022.

---

### Official Review · Reviewer_LrEX · 2025-03-14

**Overall Recommendation:** 3

**Summary:**

This paper presents TG-ECNet, a unified framework that concurrently addresses restoration and fusion of degraded visible images (affected by noise, blur, and haze) and infrared images (with stripe noise) through a task-gated router and multi-expert collaboration mechanism. The proposed integration of restoration and fusion processes enhances model robustness, demonstrating improved performance in real-world scenarios. Furthermore, the authors introduce a new benchmark dataset containing degraded multi-modal images.

**Claims And Evidence:**

Insufficient substantiation:
1. The use of only two experts in the multi-expert collaboration module raises questions about the efficacy of this core design compared to conventional multi-expert architectures.
2. No analysis is provided regarding the impact of MoE configurations on model parameters and computational overhead.

**Essential References Not Discussed:**

While the core methodology relates to dynamic networks for image fusion, the authors omit discussion of Mutual-guided Dynamic Network for Image Fusion (ACMMM 2023), a directly relevant contemporaneous work.

**Experimental Designs Or Analyses:**

Please refer to previous comments.

**Methods And Evaluation Criteria:**

Not sufficiently justified:
1. Inadequate specification of degradation parameters and data characteristics for both proposed benchmark datasets.
2. Experimental details on the detection task implementation remain unclear.

**Other Comments Or Suggestions:**

- Lines 211-213 and 213-215 contain repetitive content
- Discrepancy exists between table organization (recent methods first) and visual/metric presentation order
- Typographical error in Line 531 ("table()") requires correction

**Other Strengths And Weaknesses:**

Strengths:
- Introduction of novel benchmark dataset for degraded multi-modal restoration/fusion
- Comprehensive experiments demonstrating SOTA performance on proposed and existing datasets
- Persuasive visual comparisons showing qualitative improvements
Weaknesses:
- Limited ablation studies on critical components
- Insufficient analysis of computational efficiency

**Questions For Authors:**

Please refer to previous comments.

**Relation To Broader Scientific Literature:**

This work connects to:
All-in-one image restoration approaches.
Multi-modal image fusion methodologies.
MoE (Mixture of Experts) techniques.
Dynamic network architectures.

**Theoretical Claims:**

The paper primarily makes the following theoretical claims:
1. Multi-modal image fusion must account for potential degradations in both RGB and infrared modalities
2. Existing multi-modal fusion methods underperform when processing degraded input images
3. The task-gated router enables adaptive handling of degraded features during fusion
4. The multi-expert collaborative network achieves robust restoration and high-quality fusion simultaneously
5. A two-stage training strategy optimizes joint handling of restoration and fusion tasks

---

> ### Author Rebuttal · Authors · 2025-04-01
>
> Thanks for recognizing our contributions in this work. We will reply to your questions in order.
>
> ---
>
> ## Response to the selection of experts
> #### In our work, we select the top 6 experts from 11 experts to cope with different degradations. In our paper, there is a typo in Line 218. We utilize degradations of three types of Gaussian noise, haze, defocus blur, and stripe noise. The number of types is aligned with the number of experts we select. In [NewFig4](https://anonymous.4open.science/r/TG-ECNet/NewFig4.png), we compare the efficiency and performance of different methods and different expert selections in our method, which demonstrates that our selection is reasonable.
> ---
>
> ## Response to the impact of MoE configurations
> #### In [NewFig4](https://anonymous.4open.science/r/TG-ECNet/NewFig4.png), we have shown that our method has a medium standard of FPS and Params and achieves a great performance. As the number of experts increases, the tasks each expert needs to complete are reduced, allowing them to focus on features specific to a particular degradation. Although the computational difficulty is somewhat heightened, it increases the granularity of the experiments.
>
> ---
> ## Response to the datasets
> #### Our dataset includes six types of degradation: Noise15, Noise25, Noise50, Haze, DefocusBlur, and StripeNoise, with quantities of 726, 723, 724, 909, 3094 and 2443 respectively. Since we consider the DefocusBlur task to be more challenging and the infrared modality only involves one type of degradation StripeNoise, these two tasks have a larger volume of data. Additionally, we have created multi-degraded scenarios where a single image pair may contain 2 to 6 types of degradation, with 25 images allocated for each configuration.
> ---
>
> ## Response to the setting of the object detection
> #### For each test set containing different degradations, we split the images in an 8:2 ratio and input them into the YOLOv5 network. The training was configured with 50 epochs, an image size of [640, 640], and we calculated both AP and mAP@0.5:0.95. The entire training process was completed on a single RTX 3090 GPU.
> ---
>
> ## Response to essential references
> #### We conducted a series of experiments using MGDN [r2], with the corresponding results presented in [NewFig1](https://anonymous.4open.science/r/TG-ECNet/NewFig1.jpg), [Newfig2](https://anonymous.4open.science/r/TG-ECNet/NewFig2.png), [NewFig3](https://anonymous.4open.science/r/TG-ECNet/NewFig3.png), [NewFig4](https://anonymous.4open.science/r/TG-ECNet/NewFig4.png), [NewFig5](https://anonymous.4open.science/r/TG-ECNet/NewFig5.png) and [Newfig6](https://anonymous.4open.science/r/TG-ECNet/NewFig6.png). Our method outperformed MGDN [r2] across all tasks. We will incorporate a citation to this work in the revised manuscript.
> ---
>
> ## Response to ablation studies on critical components
> #### In [NewFig7](https://anonymous.4open.science/r/TG-ECNet/NewFig7.png), we have conducted two other ablation experiments. First, since we use a learnable mask in the fusion stage, we compare this setting with visible modality dominant(VIS Major), infrared modality dominant(IR Major), and direct VIS-IR modality addition(plus). The result demonstrates that our setting is suitable for different scenarios. Second, since we use the multi-expert collaboration module in both stages, we conduct experiments in which only one stage uses the multi-expert collaboration module. In the experiments, we demonstrate that the module is instrumental in both stages.
> ---
>
> ## Response to the analysis of computational efficiency
> #### We have responded to it above. Related results can be seen in [NewFig4](https://anonymous.4open.science/r/TG-ECNet/NewFig4.png).
> ---
>
> ## Response to other comments or suggestions
> #### 1. Lines 211-213 and 213-215 don’t contain repetitive content. Lines 211-213 denote the setting of the first stage of the network, while lines 213-215 denote the setting of the second stage of the network.
> #### 2. Thanks for this suggestion. We have adjusted the order in time order both in the table and in the visualization.
> #### 3. Thanks for correcting this typo.
> ---
> Thanks for your suggestions.
>
> [r2] Guan, et al. Mutual-guided dynamic network for image fusion. ACM MM, 2023.

---

> > ### Comment · Reviewer_LrEX · 2025-04-04
> >
> > I have carefully reviewed the authors’ rebuttal, which resolves most of my concerns. That said, I urge the authors to take the final revision seriously, addressing all reviewer comments with responsibility to the community. I also strongly encourage the authors to release the code and dataset, which would significantly enhance the reproducibility of the work and benefit the broader research community.

---

### Official Review · Reviewer_oppE · 2025-03-15

**Overall Recommendation:** 4

**Summary:**

This paper introduces TG-ECNe, a novel framework designed to address the challenges of degraded multimodal image fusion. Multimodal images, such as visible and infrared images, often suffer from degradations like noise, blur, haze, and stripe noise, which negatively impact fusion quality. TG-ECNet tackles these issues by incorporating a task-gated router that includes degradation-aware gating in the encoder and fusion-aware gating in the decoder.

**Claims And Evidence:**

Yes

**Essential References Not Discussed:**

No

**Experimental Designs Or Analyses:**

Yes

**Methods And Evaluation Criteria:**

Yes

**Other Comments Or Suggestions:**

No

**Other Strengths And Weaknesses:**

1. The experiments lack visualization results in combined degradation scenarios. Furthermore, since existing All-in-One image fusion algorithms are incapable of handling composite degradation, it is necessary to compare them with image restoration algorithms used as pre-processing steps to ensure the fairness of the experiments.

2. The article lacks detailed information about the proposed dataset, DeMMI-RF. The haze and noise scenarios do not align with real-world degradation conditions. Firstly, infrared images often suffer from varying degrees of degradation in adverse weather conditions, which the DeMMI-RF dataset does not seem to account for. Secondly, infrared images are more susceptible to noise than visible light images, yet the paper does not include experiments where both modalities are affected by noise.

3. In haze scenarios, comparisons should be made with current image fusion algorithms designed for adverse weather conditions, such as AWFusion.

4. The authors employ a multi-expert network to handle different types of degradation. However, could this lead to high computational complexity for the proposed algorithm? An analysis of computational efficiency is lacking.

5. The proposed framework does not appear to introduce any theoretical innovations at the module level. A more in-depth analysis is needed to explain why the proposed framework can effectively address challenges in complex scenarios.

6. Experiments in non-degraded scenarios are lacking, which are necessary to validate the generalization capability of the proposed algorithm.

**Questions For Authors:**

See the Weaknesses

**Relation To Broader Scientific Literature:**

No

**Theoretical Claims:**

Yes

---

> ### Author Rebuttal · Authors · 2025-04-01
>
> Thanks for recognizing our contributions in this work. We will reply to your questions in order.
>
> ---
>
> ## Response to visualization results in combined degradation scenarios
> #### In Fig.1, we’ve shown the performance of some methods like Text-if and DRMF in combined degradation scenarios. We use AdaIR(ICLR2025) as pre-processing before fusion algorithms to ensure the fairness of the experiments and show the visualization results and quantitative comparisons in [Newfig1](https://anonymous.4open.science/r/TG-ECNet/NewFig1.jpg), where our method outperforms all other methods.
> ---
>
> ## Response to the proposed dataset
> #### In our DeMMI-RF dataset, the haze scenarios are generated by the atmospheric scattering model(ASM), where we change the parameter $A$ to adjust the intensity close to real situations. From the traditional ASM, we know:
>
> #### $$J(x)=\frac{1}{t(x)}I(x)-A\frac{1}{t(x)}+b$$
>
> #### Where $t(x)$ indicates image with haze, $J(x)$ indicates image without haze, $A$ indicates atmospheric light intensity. In our work, we set $A$ as $\overline{A}=MEAN(J(x))$, which avoids the image being too bright and makes it close to reality.
> #### The noise scenarios are aligned with other Restoration Model settings.
> #### Besides, we utilize real-world data from the AWMM dataset[r1] to validate the robustness of our method in [Newfig2](https://anonymous.4open.science/r/TG-ECNet/NewFig2.png). In [Newfig2](https://anonymous.4open.science/r/TG-ECNet/NewFig2.png), we show the visualization comparison among related works and ours. The dataset consists of images with snow and haze. In our experiment, the results of our method have improved clarity and contrast.
> ---
>
> ## Response to the methods designed for adverse weather conditions
> #### Comparisons with AWFusion [r1] have been made in [NewFig1](https://anonymous.4open.science/r/TG-ECNet/NewFig1.jpg), [Newfig2](https://anonymous.4open.science/r/TG-ECNet/NewFig2.png), [NewFig3](https://anonymous.4open.science/r/TG-ECNet/NewFig3.png), [NewFig4](https://anonymous.4open.science/r/TG-ECNet/NewFig4.png), [NewFig5](https://anonymous.4open.science/r/TG-ECNet/NewFig5.png) and [Newfig6](https://anonymous.4open.science/r/TG-ECNet/NewFig6.png). Our method demonstrates superior performance across various scenarios. We will duly incorporate a citation to this work in the revised manuscript.
> ---
>
> ## Response to the computational efficiency
> #### Although our multi-expert network inevitably increases the computational complexity, our method has a smaller parameters than some of the other methods. As for the number of experts, we show the quantitative comparisons between related works and other MoE settings in our work on the performance and efficiency in [NewFig4](https://anonymous.4open.science/r/TG-ECNet/NewFig4.png), which shows our choice is best when considering the efficiency and performance. Compared with other methods, our method has a medium cost but shows a good performance. In the selection of the number of experts, our selection shows the best performance.
> ---
>
> ## Response to innovations
> #### The traditional two-stage approach (image restoration followed by fusion) may be contradictory. Image restoration tends to eliminate noisy information while image fusion tends to integrate more valid information, but the restoration model may eliminate information of interest for image fusion in the first stage, leading to sub-optimal fusion performance. In our work, the proposed unified framework realizes the divide-and-conquer treatment of different degradation tasks by arranging the corresponding experts to process dynamically through Degradation-Aware Gating performing task routing, while the multi-expert system guided by Fusion-Aware Gating dynamically equilibrates the degree of information retention between the fusion and restoration tasks to realize the better restoration and fusion results. Our framework reduces the contradiction between the restoration and fusion tasks and minimizes the information loss in the cascaded structure. Also, the structure doesn’t need text guidance, which is a must for Text-if and is more adaptive for multitasks than DRMF. Such a network can recognize different degradations and extract effective features.
> ---
>
> ## Response to experiments in non-degraded scenarios
> #### We’ve shown the results in non-degraded scenarios in [NewFig5](https://anonymous.4open.science/r/TG-ECNet/NewFig5.png). From the results, we can know that our method is immune to the loss of features between cascaded networks.
> ---
> Thanks for your suggestions.
>
> [r1] Li, et al. AWFusion. arXiv:2402.02090,2024.

---

### Decision · Program_Chairs · 2025-05-01

**Decision:**

Accept (poster)

**Comment:**

The final scores for this submission are 2 accepts and 2 weak accepts, showing a consistent and generally positive assessment. All reviewers chose to raise their ratings after the rebuttal phase, with most indicating that the authors had successfully addressed their initial concerns. In light of these considerations, I believe this work, with some further refinement, is capable of meeting the standards of ICML 2025, and I am therefore inclined to recommend acceptance.